# QPM: Discrete Optimization for Globally Interpretable Image Classification

**Thomas Norrenbrock, Timo Kaiser & Bodo Rosenhahn**
Institute for Information Processing (tnt)
L3S - Leibniz Universität Hannover, Germany
`{norrenbr,kaiser,rosenhahn}@tnt.uni-hannover.de`

**Sovan Biswas & Ramesh Manuvinakurike**
Intel Labs, USA
`{sovan.biswas,ramesh.manuvinakurike}@intel.com`

## Abstract

Understanding the classifications of deep neural networks, *e.g.* used in safety-critical situations, is becoming increasingly important. While recent models can locally explain a single decision, to provide a faithful global explanation about an accurate model's general behavior is a more challenging open task. Towards that goal, we introduce the Quadratic Programming Enhanced Model (QPM), which learns globally interpretable class representations. QPM represents every class with a binary assignment of very few, typically 5, features, that are also assigned to other classes, ensuring easily comparable contrastive class representations. This compact binary assignment is found using discrete optimization based on predefined similarity measures and interpretability constraints. The resulting optimal assignment is used to fine-tune the diverse features, so that each of them becomes the shared general concept between the assigned classes. Extensive evaluations show that QPM delivers unprecedented global interpretability across small and large-scale datasets while setting the state of the art for the accuracy of interpretable models.

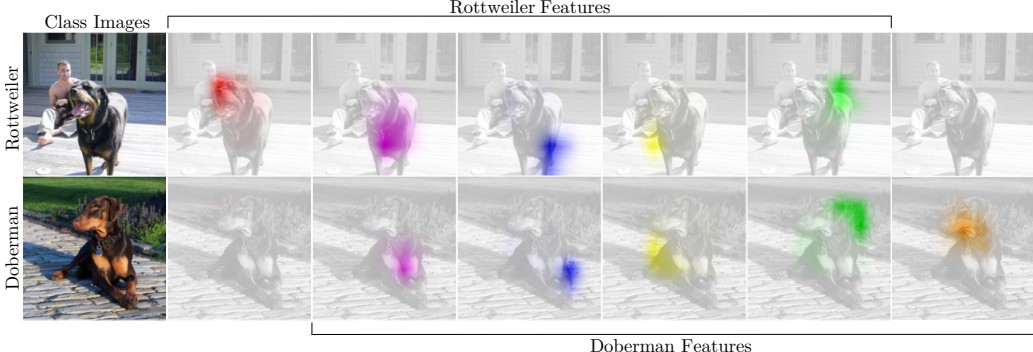

Figure 1: Faithful global interpretability of our QPM: Without any additional supervision, QPM learns to represent Rottweiler and Doberman using 5 diverse and general features. QPM faithfully explains that it differentiates them exclusively via their visibly distinct head.

## 1 Introduction

Deep Learning has made remarkable advances in various fields, such as image classification, segmentation or generation (Krizhevsky et al., 2012; Kirillov et al., 2023; Rombach et al., 2021; Ramesh et al., 2022). For high-stakes decisions, *e.g.* applying image classification in the medical domain, legislation moves towards requiring a certain level of interpretability (Veale & Zuiderveen Borgesius, 2021), whose measurement is a fairly open task on its own. However, some desirable and measurable

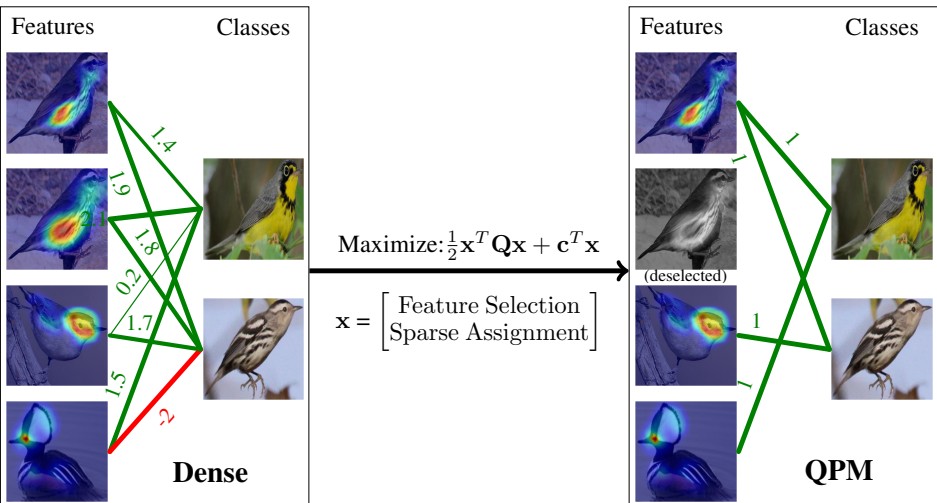

Figure 2: Exemplary Application of the QP to a dense model with just 4 features and 2 classes with the aim of selecting 3 features and assigning 2 per class. The different weights are indicated by thickness of connection and color indicates sign. The result is a binary assignment of selected features to classes. Typical values for Resnet50 (He et al., 2016) on CUB-2011 (Wah et al., 2011) are selecting 50 features out of 2048 and assigning 5 to each of the 200 classes.

qualities of explanations have been identified (Miller, 2019). Human-friendly explanations should be contrastive (Lipton, 1990), diverse (Alvarez Melis & Jaakkola, 2018), general and compact (Read & Marcus-Newhall, 1993). As humans can consider $7 \pm 2$ cognitive aspects at once (Miller, 1956), an explanation size of up to $5$ is desirable. Additionally, an explanation should faithfully explain the model, which is where many post-hoc methods fail (Kindermans et al., 2019; Adebayo et al., 2018; Daras & Dimakis, 2022). Therefore, we focus on models that are interpretable by design with built-in faithful explanations.

Previous works, such as SENN (Alvarez Melis & Jaakkola, 2018), *Q-SENN* (Norrenbrock et al., 2024), *Concept Bottleneck Model* (CBM) (Koh et al., 2020), *Label-free CBM* (Oikarinen et al., 2023), *PIP-Net* (Nauta et al., 2023), *ProtoPool* (Rymarczyk et al., 2022), *ProtoTree* (Nauta et al., 2021), *ProtoPNet* (Chen et al., 2019) or the *SLDD-Model* (Norrenbrock et al., 2022) rely on combining understandable features in an interpretable manner. However, while most models can offer convincing *local* explanations for a single decision, they struggle with the *global* explanation of their behavior in general. Some models with global interpretability do not show competitive accuracy (Oikarinen et al., 2023; Koh et al., 2020) and it is debated (Molnar, 2020), if ensembles of very deep decision tress (Nauta et al., 2021) or dense high-dimensional linear layers (Koh et al., 2020; Rymarczyk et al., 2022; Alvarez Melis & Jaakkola, 2018) are truly intrinsically interpretable as they lack desired qualities like compactness. For that reason *PIP-Net* focuses on learning sparse class representations. These representations lie in a high dimensional feature space, which causes *PIP-Net*'s features to be connected to very few or only one class each. This leads to the emergence of features that are already detecting the class and no general concept. The sparse representations of *PIP-Net* thus have no interpretable meaning, as classes are represented with themselves. To alleviate that issue, the *SLDD-Model* and *Q-SENN* reduce both dimensions of compactness: They not only reduce the number of features per class $n_{wc}$, which in isolation leads to class-specific features but also the number of features in total $n_f^*$ to be significantly below the number of classes $n_c$. That causes each of the fewer features to be assigned to multiple classes, which prevents the emergence of class detectors. However, these models still have shortcomings when it comes to global interpretability. Their class representations are real-valued, or ternary for *Q-SENN*, include a bias, and are composed of a varying number of features. Therefore, the global class explanations are hardly comparable or contrastive.

In this work, we introduce the Quadratic Programming Enhanced Model (QPM) that offers interpretable class representations and sets a new state of the art for the accuracy of compactness-based interpretable models. It represents every class with the binary assignment of a low user defined number of features $n_{wc}$, which themselves are contrastive, general and diverse. We typically choose $5$, in line with previous work (Norrenbrock et al., 2024; 2022), to accommodate for human limitations (Miller, 1956). As shown in fig. 1, QPM offers built-in faithful global explanations for classes

Table 1: Properties of class representation for class $i$, $y_i = \mathbf{w_i}\mathbf{f} + b_i$, for CUB-2011: Only QPM represents each of its classes with the binary assignment of a fixed number of general features (quantified in table 3) and no class Bias. Therefore, classes can also be represented as set of 5 feature indices $S_i$, $y_i = \sum_{j \in S_i} f_j$. These contrastive class explanations enable faithful global interpretability. If applicable, all methods are configured to $n_{\mathrm{wc}} = 5$ and $n_f^* = 50$.

| Method | Size of $\mathbf{w_i}$ | Equal Class Sparsity | No Class Bias | Contrastive Representation | |
|---|---|---|---|---|---|
| Baseline Resnet50 | $\mathbf{w_i} \in \mathbb{R}^{2048}$ | ✓ | ✗ | | ✗ |
| glm-saga$_5$ | $\mathbf{w_i} \in \mathbb{R}^{809}$ | ✗ | ✗ | | ✗ |
| PIP-Net | $\mathbf{w_i} \in \mathbb{R}^{731}$ | ✗ | ✓ | | ✗ |
| ProtoPool | $\mathbf{w_i} \in \mathbb{R}^{202}$ | ✓ | ✓ | | ✗ |
| SLDD-Model | $\mathbf{w_i} \in \mathbb{R}^{50}$ | ✗ | ✗ | | ✗ |
| Q-SENN | $\mathbf{w_i} \in \{-\alpha, 0, \alpha\}^{50}$ | ✗ | ✗ | | ✗ |
| QPM (Ours) | $\mathbf{w_i} \in \{0, 1\}^{50}$ | ✓ | ✓ | $S_i \in \{1, \ldots, 50\}^5$ | ✓ |

and enables the intuitive comparison of different learned class representations. These easy comparisons between compact binary class representations even enable reasoning about the differentiating feature between the classes, like the head in fig. 1. The improvements in faithful global interpretability of class representations are summarized in table 1.

The crucial step in training a QPM is solving a binary QP, applied to a dense black-box model, which jointly finds an optimal solution to both the selection of a reduced subset of the model's features and the sparse assignment between the features and classes, as shown in fig. 2. It maximizes the similarity between features and their assigned classes, while minimizing the similarity of jointly selected features. Further, the linear term can steer the selection towards desired biases, while the desired interpretability is incorporated via constraints. This optimal solution is then fixed for the following fine-tuning during which the features adapt to their assigned classes. As every class is assigned to the same number of features, each of the features detects shared general concepts between its assigned classes instead of also detecting the entire class. This leads to state-of-the-art accuracy. Finally, the assignments are not maximizing inter-class distance, resulting in more similar representations for similar classes and a form of structural grounding. Code: `https://github.com/ThomasNorr/QPM`

Our main **contributions** are as follows:

- We propose the Quadratic Programming Enhanced Model (QPM), which incorporates an optimal feature selection and their binary assignment of a few, *e.g.* 5 features per class. It is found by formulating the quadratic problem and solving it optimally.

- We demonstrate improvements in accuracy, compactness and structural grounding of QPM on multiple benchmark datasets and architectures for image classification, including ImageNet-1K (Russakovsky et al., 2015). Due to optimally using the given capacity, QPM sets the new state of the art for compactness-based globally interpretable models.

- We show that the learned features exhibit several desired quantifiable properties, such as contrastiveness, generality and diversity, and can be steered towards user-defined criteria.

- Representing classes as a contrastable compact set of these general features makes QPM faithfully globally interpretable, while further closing the accuracy gap to black-box models.

## 2 RELATED WORK

Research towards Interpretable machine learning includes the direct design of models providing interpretability by themselves (Alvarez Melis & Jaakkola, 2018; Sawada & Nakamura, 2022; Norrenbrock et al., 2022; Nauta et al., 2023; 2021; Rymarczyk et al., 2022; Zarlenga et al., 2022; Marconato et al., 2022; Koh et al., 2020; Rymarczyk et al., 2021; Chen et al., 2019) or to find post-hoc methods which aim to explain the decision process or single features of the model (Kim et al., 2018; Bau et al., 2017; McGrath et al., 2022; Fel et al., 2023; Yuksekgonul et al., 2022; Kalibhat et al., 2023; Oikarinen & Weng, 2023). As our method is designed to find a compact set of human-understandable features, our work can be assigned to the former type, which we focus on within this section. However, the alignment of the learned features of our proposed QPM with human attributes can be guided by the post-hoc methods. When considering the interpretability of a model, a distinction is made between local interpretability, which refers to the explanation of a single decision, and global interpretability,

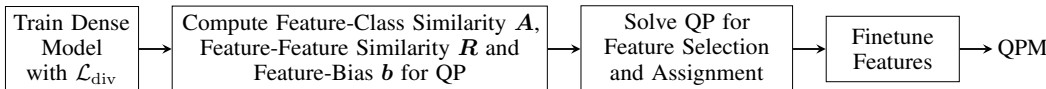

Figure 3: Overview of our proposed pipeline to construct a QPM

which describes the holistic behavior of the model over the entirety of a dataset (Molnar, 2020). For local interpretability, *B-Cos Networks* (Böhle et al., 2023) already offer faithful explanations in the form of saliency maps. Therefore, this work focuses on the more challenging global interpretability, which also improves local interpretability. In the social sciences (Miller, 2019), human-friendly explanations are contrastive (Lipton, 1990), concise and general (Read & Marcus-Newhall, 1993). Further, SENN (Alvarez Melis & Jaakkola, 2018) describes diversity and grounding as desirable attributes for features of an interpretable model. Grounding refers to the alignability with any human concept and is very difficult to quantify, as one would need a full dataset of potentially learned concepts. Problematically, deep neural networks typically exhibit superposition and polysemantic neurons (Scherlis et al., 2022; Elhage et al., 2022; Templeton, 2024), which is why we focus on more clearly quantifiable aspects in this work.

Models such as *Prototree* (Nauta et al., 2021), *ProtoPNet* (Chen et al., 2019), *ProtoPShare* (Rymarczyk et al., 2021), *ProtoPool*, and *PIP-Net* aim to learn prototypes from data by employing deep feature extractors. These prototypes' similarities are subsequently integrated into interpretable models. However, the extent of their interpretability remains debatable, as Kim et al. (2022) and Hoffmann et al. (2021) reveal a gap between human and computed similarities. Similar to this work, *PIP-Net* also aims for compactness via sparse weights in the final decision layer. However, they apply a local optimization that aims for sparsity solely, resulting in a big set of used features with many of them being class-specific. Norrenbrock et al. (2022; 2024) additionally select a compact feature set for their *SLDD-Model* and *Q-SENN*, where a class is to be related to only a few features. Their diversity is ensured through the Feature Diversity Loss $\mathcal{L}_{\mathrm{div}}$, which incurs a higher cost when highly activated and weighted features localize on the same region. For both feature selection and the computation of the sparse layer, glm-saga (Wong et al., 2021) is used. It locally and iteratively optimizes the problem, leading to a suboptimal feature selection and continuous weights. In contrast, our global optimization with user-defined steerable criteria jointly finds an optimal selection of the required number of features and computes their binary assignments. This leads to a more effective use of the allocated capacity and built-in easily interpretable class representations for global interpretability. Another line of research is based on the *Concept Bottleneck Model* (CBM) which initially predicts the labeled concepts within a given dataset and subsequently leverages a basic model to predict the target category based on these identified concepts. This approach remains an area of active exploration and development (Sawada & Nakamura, 2022; Zarlenga et al., 2022; Marconato et al., 2022; Oikarinen et al., 2023), but is limited by the annotations, or in case of the *Label-free CBM* by the vision-language model, resulting in subpar accuracy and compactness. Finally, Rosenhahn (2023) applies discrete optimization to obtain sparse neural networks (Glandorf* et al., 2023).

## 3 METHOD

Our proposed QPM is designed for the interpretable classification of an image as a class $c \in \{c_1, c_2, \ldots, c_{n_c}\}$. The QPM uses a deep feature extractor $\Phi$ to compute feature maps $\boldsymbol{M} \in \mathbb{R}^{n_f^* \times w_M \times h_M}$ of width $w_M$ and height $h_M$ and averages them into a feature vector $\boldsymbol{f}^* \in \mathbb{R}^{n_f^*}$. The classification result $\boldsymbol{y} \in \mathbb{R}^{n_c}$ of the QPM is the matrix multiplication between the sparse binary matrix $\boldsymbol{W}^* \in \{0,1\}^{n_c \times n_f^*}$ and the features $\boldsymbol{f}^*$ formalized as $\boldsymbol{y} = \boldsymbol{W}^* \boldsymbol{f}^*$.

The pipeline of our proposed method is shown in fig. 3 and is motivated by (Norrenbrock et al., 2022; 2024), following their presentation and notation. It starts with training a conventional black-box model with initially $n_f$ features using the feature diversity loss $\mathcal{L}_{\mathrm{div}}$ (Norrenbrock et al., 2022), as a high diversity of features is desired for interpretable models. A detailed explanation of $\mathcal{L}_{\mathrm{div}}$ is included in appendix M. Using the black-box model as starting point, we aim to find a selection of $n_f^*$ out of the initial $n_f$ features and their sparse binary assignment $\boldsymbol{W}^*$ to the classes to enable downstream interpretability. The feature extractor $\Phi$ is then fine-tuned with this solution fixed, so that the features adapt to the sparse solution and become a shared concept of the assigned classes. This is encouraged through selecting fewer features than there are classes, $n_f^* < n_c$, and representing every class with the same number $n_{\mathrm{wc}}$, typically 5, of features. Using the same number of features for every class is beneficial for the interpretability in multiple ways. The class representations do not

need a bias and can be contrasted as $S_i \in \{1, \ldots, n_f^*\}^{n_{wc}}$, while the composing features can focus on detecting general concepts. Since we aim to optimize binary variables under constraints with a clear objective, we can formulate it as a discrete optimization problem to get the optimal solution. As indicated in fig. 2, we define the constants $\boldsymbol{A}$, $\boldsymbol{R}$ and $\boldsymbol{b}$ of the resulting QP so that in the global optimum different ($\boldsymbol{R}$), localized ($\boldsymbol{b}$) features are selected and assigned to classes for which they have high predictive power ($\boldsymbol{A}$). These fixed simple binary class representations then lead to the emergence of interpretable features during fine-tuning. How the quadratic problem with $\boldsymbol{A}$, $\boldsymbol{R}$ and $\boldsymbol{b}$ is formulated to ensure this goal is discussed in the following sections.

## 3.1 QUADRATIC PROBLEM

We consider the problem of selecting the $n_f^*$ out of $n_f$ features and assigning them to the classes as a binary quadratic problem, that can be solved globally optimal. Specifically, the feature selection $\boldsymbol{s} \in \{0,1\}^{n_f}$ and assignment between features and classes $\boldsymbol{W} \in \{0,1\}^{n_c \times n_f}$ are jointly optimized, with $\boldsymbol{W}^*$ being $\boldsymbol{W}$ for the selected features. Given a similarity matrix $\boldsymbol{A} \in \mathbb{R}^{n_c \times n_f}$ the main objective is to maximize the similarity $Z_A$ between the selected features and their assigned classes

$$Z_A = \sum_{c=1}^{n_c} (\boldsymbol{a}_c \circ \boldsymbol{w}_c)^T \boldsymbol{s} \tag{1}$$

with $\circ$ indicating the Hadamard product. Here, $\boldsymbol{s}$ indicates whether a feature is selected and $\boldsymbol{W}$ describes if a feature is assigned to the class. Note that we use $c$ to index classes and $d$ for features. The sparsity and low-dimensionality are formulated as constraints for the optimization:

$$\sum_{d=1}^{n_f} s_d = n_f^* \tag{2}$$

$$\sum_{d=1}^{n_f} w_{c,d} s_d = n_{wc} \quad \forall c \in \{1, \ldots, n_c\} \tag{3}$$

To allow the QPM the differentiation between all classes and enable effective fine-tuning, we additionally add constraints that no two classes are assigned to the same set of features:

$$(\boldsymbol{w}_c \circ \boldsymbol{w}_{c'})^T \boldsymbol{s} < n_{wc} \quad \forall c, c' \in \{1, \ldots, n_c\} \tag{4}$$

Note that the constraints in eqs. (3) and (4) technically define a quadratically constrained quadratic program (QCQP). To make the QCQP computationally tractable, the constraints are relaxed and added iteratively for classes that violate the constraints. The efficient implementation is discussed in detail in section 4.1.1. The general formulation of the problem allows us to add further nuance to the optimization and include more desiderata. Since a high representational capacity is desired for the selected features, the cross-feature similarity matrix $\boldsymbol{R} \in \mathbb{R}^{n_f \times n_f}$ is incorporated to reduce the similarity between the selected features:

$$Z_R = -\boldsymbol{s}^T \boldsymbol{R} \boldsymbol{s} \tag{5}$$

Additionally, the selection of specific features can be guided via a selection bias $\boldsymbol{b} \in \mathbb{R}^{n_f}$

$$Z_B = \boldsymbol{b}^T \boldsymbol{s}, \tag{6}$$

where a higher value $\boldsymbol{b}_i$ leads to a preferred selection of the feature $i$. The combination of all these objectives leads to:

$$\max_{\boldsymbol{W}, \boldsymbol{s}} Z = \max_{\boldsymbol{W}, \boldsymbol{s}} Z_A + Z_R + Z_B \tag{7}$$

The formulation in standard form for quadratic problems $\frac{1}{2}\mathbf{x}^T \mathbf{Q} \mathbf{x} + \mathbf{c}^T \mathbf{x}$ with $\mathbf{Q}$ capturing the quadratic terms $Z_A$ and $Z_R$, and $\mathbf{c}$ incorporating the linear term $Z_B$ is included in appendix O.

## 3.2 CLASS-FEATURE SIMILARITY

The class-feature similarity matrix $\boldsymbol{A}$ with entries $a_{c,d}$ should reflect how beneficial the assignment of feature $d$ to class $c$ is for the classifier. As every feature gets assigned to multiple classes, which themselves become assigned to multiple features, the metric should focus on a robust positive relation between the activation and likelihood of a sample being of the respective class. This is captured by the Pearson correlation coefficient $a_{c,d}$ between the feature distribution $\boldsymbol{f}_{:,d}$ and the label vector $\boldsymbol{l}^c \in \{0,1\}^{n_T}$, in which for all $n_T$ training images a 1 indicates the label being $c$.

### 3.3 Feature-Feature Similarity

Just maximizing eq. (1) can lead to very similar features being selected which is neither beneficial for interpretability nor for accuracy as representational capacity is lost and multiple features develop towards the same concept during fine-tuning. To prevent this, selecting similar features in $\boldsymbol{A}$ should be penalized in the objective. We choose the cosine similarity between the class similarities of two features $d \neq d'$ in $\boldsymbol{A}$ for $\boldsymbol{R}$ with $r_{d,d'} = \text{ReLU}\left(\frac{\boldsymbol{a}_{:,d}^T \boldsymbol{a}_{:,d'}}{|\boldsymbol{a}_{:,d}||\boldsymbol{a}_{:,d'}|}\right)$, using ReLU to focus on preventing redundant features and $r_{d,d'} = 0$ for $d = d'$. As we are only interested in preventing the selection of highly similar features, we can clip all entries in $\boldsymbol{R}$ below an $\epsilon$ to 0 to enable a fast solving of the QP. The details are discussed in section 4.1.1.

### 3.4 Feature-Bias

The Feature-Bias $\boldsymbol{b}$ describes the benefit of selecting each feature. This can be used to steer the model towards specific desiderata. As diversity is generally preferred (Norrenbrock et al., 2022; Alvarez Melis & Jaakkola, 2018) for interpretable models, a bias towards more local features is used,

$$b_d = \frac{1}{n_T \sum_j f_{j,d}} \sum_{j=1}^{n_T} max(\boldsymbol{S}_j^d) f_{j,d} \quad . \tag{8}$$

Here $\boldsymbol{S}_j^d$ is the softmax over the spatial dimensions of the $d$-th feature map for the image $j$. Scaling the feature bias by their activation leads to the selection of features that are more localized when their activation is high. Alternatively, the bias can be used to steer the selection towards other criteria the practitioner might identify as relevant, which we demonstrate in the appendix. We center $\boldsymbol{b}$ and scale the maximum absolute value to be $\lambda$, whose strength defines the priority put on the bias.

## 4 Experiments

Following prototype-based methods we applied our method to CUB-2011 (Wah et al., 2011) and Stanford Cars (Krause et al., 2013). To showcase QPM's broad applicability, we also include results on the large-scale dataset ImageNet-1K (Russakovsky et al., 2015), to which most interpretable methods are not applicable. Notably, CUB-2011 contains annotations of human concepts which we use to measure Structural Grounding. An overview of the used datasets is shown in Suppl. table 5. As our method is independent of the used backbone, we evaluated it across various architectures, but focus on Resnet50 (He et al., 2016) in this paper. Similar results on Resnet34, Inception-v3 (Szegedy et al., 2016) and Swin Transformer (Liu et al., 2021), as well as detailed results with standard deviations, are included in Suppl. appendix L. We do not apply our method to other interpretable models like *PIP-Net* (Nauta et al., 2023), as QPM is an alternative way of inducing compactness and the features of *PIP-Net* are not general, thus ill-suited for a broad assignment.

### 4.1 Implementation Details

We generally followed *PIP-Net* for the data preparation. Specifically, the images are first cropped to the ground truth bounding box for CUB-2011 and TravelingBirds (Koh et al., 2020). For all datasets, the images are resized to $224 \times 224$. Following *PIP-Net*, *TrivialAugment* (Müller & Hutter, 2021) is used and the strides of ResNets are also set to 1 to obtain more fine-grained feature maps. The remaining parameters, including dense training for 150 epochs on fine-grained datasets and directly using the pretrained model on ImageNet-1K with subsequent 40 epochs of fine-tuning, mirror the *SLDD-Model* and are described in appendix C. Note that QPM is trained more efficiently than *Q-SENN*, as it does not use multiple training iterations during fine-tuning. We set $n_{\text{wc}} = 5$ and $n_f^* = 50$ for QPM, unless stated otherwise. We demonstrate the impact of changing the parameters in the ablation studies but choose these, as it is in line with prior literature (Norrenbrock et al., 2024; 2022), $n_f^* < n_c$, and it enables sufficiently compact explanations (Miller, 1956). The shown results, *e.g.* tables 2 and 3, are the mean across 5 seeds, with the exception of 3 for ImageNet-1K, *PIP-Net* and *ProtoPool*. For comparison, all models are exclusively pretrained on ImageNet-1K. This change did affect *ProtoPool*, but even with iNaturalist (Van Horn et al., 2018) pretraining, we could not reproduce the reported results by Rymarczyk et al. (2022).

#### 4.1.1 QUADRATIC PROBLEM

This section presents details on how the described quadratic problem with eq. (7) as objective is solved using *Gurobi* (Gurobi Optimization, LLC, 2023). We incorporated deduplication and the assignment of an equal number of features to all classes of eqs. (3) and (4) using an iterative approach with relaxed constraints. Specifically, the model is optimized without these constraints, but instead $\mathbf{1}^T \boldsymbol{W} \boldsymbol{s} = n_{\text{wc}} n_c$. Then, after each iteration, all violated constraints are added to the model, but only limited to a running set of features $\Gamma \in \{0, 1\}^{n_f}$, which gets extended during the iteration. Next to the features, we also maintain a set of classes $C_{\text{duplicates}}$ that were equal at one iteration and classes $C_{\text{sparse}}$ that ever had too few features assigned. Instead of eqs. (3) and (4) the relaxed constraints

$$\boldsymbol{w}_{c,\Gamma}^T \boldsymbol{s}_\Gamma \geq n_{\text{wc}} \quad \forall c \in C_{\text{sparse}} \tag{9}$$

$$(\boldsymbol{w}_c \circ \boldsymbol{w}_{c'})_\Gamma^T \boldsymbol{s}_\Gamma < n_{\text{wc}} \quad \forall c, c' \in C_{\text{duplicates}} \tag{10}$$

are added, where $\boldsymbol{W}_{c,\Gamma}$ describes indexing $\boldsymbol{W}_c$ where $\Gamma = 1$. Additionally, we set the start solution for the next optimization to a good, usually optimal, feasible solution for the currently selected set of features. As we need multiple iterations to enforce all constraints, we limit the time spent on one iteration to 3 hours and set the gap to optimality to $10^{-4}$. In our experiments, the global optimum for the relaxed problem is usually found in less than 4 hours for fine-grained datasets, and roughly 11 hours for ImageNet-1K using a CPU like *EPYC 72F3*. While eq. (9) changes the desired optimization problem, the resulting objective is very close (achievable gap of less than 1%) to the global optimum, which is infeasible to compute and does not lead to an improved model. The experiments to verify this claim are included in Suppl. appendix N. Finally, alongside our experiments, previous work (Hornakova et al., 2021) shows that the exact global optimum is not always preferred for relevant metrics. To make the relative weighting of the multiple objectives $Z_A$, $Z_R$ and $Z_B$ easier, $\boldsymbol{A}$ is scaled with $n_c$ and $n_{\text{wc}}$ to have a maximum of 1 for $n_c = 200$ and $n_{\text{wc}} = 5$. Since $n_f^*$ features need to be chosen, all entries below $\epsilon$ in $\boldsymbol{R}$ are set to 0, where $\epsilon$ is the highest value, for which there still exists a selection with $Z_R = 0$. This is equivalent to finding the maximal $\epsilon$ for which the graph described by $\boldsymbol{G}$ with

$$g_{d,d'} = \begin{cases} 0 & \text{if} \quad r_{d,d'} \geq \epsilon \\ 1 & \text{else,} \end{cases} \tag{11}$$

has a maximum clique of size $n_f^*$. We used approximations (Pattabiraman et al., 2015; Boppana & Halldórsson, 1992) and a sufficiently sized approximated maximum clique as the start value for $\boldsymbol{s}$. Additionally, the remaining nonzero values in $\boldsymbol{R}$ are scaled to have a maximum of 1. For scaling the bias $\boldsymbol{b}$, we clipped outliers, centered the remaining values around 0 and scaled the maximum absolute value to be $\lambda = \frac{1}{\sqrt{10}}$, which is empirically found.

### 4.2 METRICS

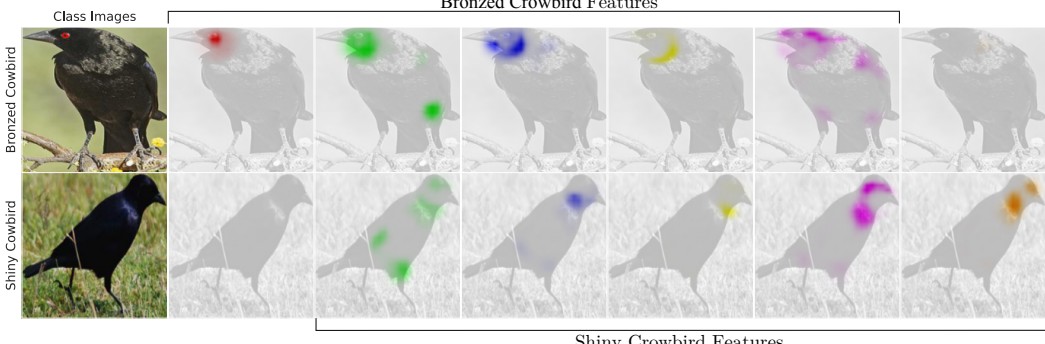

Figure 4: Contrastive faithful class explanations for QPM trained on CUB-2011: Without any additional supervision, QPM learns to differentiate Shiny and Bronzed Cowbird ($\Psi^{gt} = 0.97$) using the red eye just like humans do, as the annotations in CUB-2011 or the screenshot in fig. 24 show.

Following *PIP-Net* as recent work, we evaluate the accuracy and compactness, measured as number of total features $n_f^*$ and number of features per class $n_{\text{wc}}$. Additionally, our QPM learns interpretable

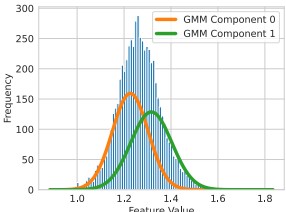 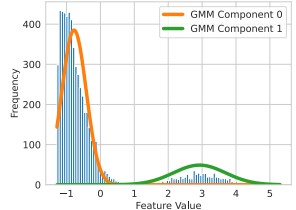

(a) Baseline Resnet50 with Contrastiveness $= 41.8\%$      (b) QPM with Contrastiveness $= 99.9\%$

Figure 5: Extreme Examples for feature distributions and their Contrastiveness on CUB-2011.

class representations, summarized in table 1, that are composed of features. As discussed in section 2, diversity, contrastiveness, generality and grounding are desired aspects of explanations. While we believe that our sparse binary assignment is very well suited for a detailed analysis and alignment of the learned features, as it likely prohibits superposition, polysemantic neurons are still likely to occur and hard to measure for QPM and all end-to-end trained interpretable models. Therefore, we omit measuring the grounding of features and instead focus on contrastive, general and diverse as desirable *and* quantifiable qualities of features as building blocks of our interpretable class representations, whose Structural Grounding we estimate using the attributes contained in CUB-2011. Specifically, every class $c$ in CUB-2011 is annotated with a vector $\boldsymbol{a}_c \in [0,1]^{312}$, where $a_{c,j}$ indicates the fraction of images with label $c$, in which a human perceives the attribute $j$ to be present. With these vectors, we compute the ground truth structural class similarity $\boldsymbol{\Psi}^{gt} \in [0,1]^{n_c \times n_c}$ with $\psi_{c,c'}^{gt}$ being the cosine similarity between $\boldsymbol{a}_c$ and $\boldsymbol{a}_{c'}$. Similarly, $\boldsymbol{\Psi}^{Model} \in [-1,1]^{n_c \times n_c}$ is based on the class vectors in the interpretable classification layer. We then report the similarity for the top 25 most similar unique pairs of classes $C_{\text{Sim}}$ in reality

$$\text{StructuralGrounding} = \frac{\sum_{c,c' \in C_{\text{Sim}}} \psi_{c,c'}^{Model}}{\sum_{c,c' \in C_{\text{Sim}}} \psi_{c,c'}^{gt}}. \tag{12}$$

Models with high Structural Grounding offer an interpretable human-like class-similarity, *e.g.* using the apparently different head to differentiate between Rottweiler and Doberman in fig. 1 or differentiating shiny and bronzed cowbird by its only separating attribute, shown in fig. 4.

To measure the contrastiveness of features, a Gaussian mixture model with two components is fit to every feature distribution $\boldsymbol{f}_{:,d}$, resulting in the normal distributions $\mathcal{N}_1^d$ and $\mathcal{N}_2^d$, visualized in fig. 5. We then compute the *Contrastiveness* as average of all features using the overlap (Inman & Bradley, 1989) between the two distributions:

$$\text{Contrastiveness} = \sum_{d=1}^{n_f^*} 1 - \text{Overlap}(\mathcal{N}_1^d, \mathcal{N}_2^d), \tag{13}$$

as bi-modal contrastive features can be represented by two non-overlapping distributions. The binary quality of the features is also indicated in figs. 1 and 4, as the features are normed per column.

Additionally, the features should capture a general concept, instead of a class-specific one. This can be measured via the *Class-Independence* $\tau$:

$$\tau = 1 - \frac{1}{n_f^*} \sum_{d=1}^{n_f^*} \max_c \frac{\sum_{j=1}^{n_T} l_j^c (f_{j,d} - \min \boldsymbol{f}_{:,d})}{\sum_{j=1}^{n_T} (f_{j,d} - \min \boldsymbol{f}_{:,d})} \tag{14}$$

It measures which fraction of the zero-based feature activation across the entire dataset is not focussed on the most related class. A model with high Class-Independence has features that recognize a shared concept for multiple classes, like the 4 central features in figs. 1 and 4. Notably, as opposed to Dependence (Norrenbrock et al., 2024), Class-Independence can capture the assignment of multiple class detectors to the same class.

For measuring the spatial diversity of the features, diversity@5 (Norrenbrock et al., 2022) has been proposed. The diversity@5 however suffers from the non-linear behavior of the softmax, resulting in

Table 2: Comparison on compactness and accuracy with Resnet50: QPM shows increased accuracy and compactness. The compactness-accuracy trade-off is shown in fig. 7. Among more interpretable models, the best result is marked in bold, second best underlined.

| Method | Accuracy ↑ | | | Total Features ↓ | | | Features / Class ↓ | | |
|---|---|---|---|---|---|---|---|---|---|
| | CUB | CARS | INET | CUB | CARS | INET | CUB | CARS | INET |
| Baseline Resnet50 | 86.6 | 92.1 | 76.1 | 2048 | 2048 | 2048 | 2048 | 2048 | 2048 |
| glm-saga$_5$ | 78.0 | 86.8 | 58.0 | 809 | 807 | 1627 | **5** | **5** | **5** |
| PIP-Net | 82.0 | 86.5 | - | 731 | 669 | - | 12 | 11 | - |
| ProtoPool | 79.4 | 87.5 | - | 202 | 195 | - | 202 | 195 | - |
| SLDD-Model | 84.5 | 91.1 | 72.7 | **50** | **50** | **50** | **5** | **5** | **5** |
| Q-SENN | 84.7 | 91.5 | **74.3** | **50** | **50** | **50** | **5** | **5** | **5** |
| QPM (Ours) | **85.1** | **91.8** | 74.2 | **50** | **50** | **50** | **5** | **5** | **5** |

Table 3: Comparison on Interpretability metrics with Resnet50. Due to required annotations, Structural Grounding (abbreviated SG) can only be computed for CUB-2011.

| Method | SID@5 ↑ | | | Class-Independence ↑ | | | Contrastiveness ↑ | | | SG ↑ |
|---|---|---|---|---|---|---|---|---|---|---|
| | CUB | CARS | INET | CUB | CARS | INET | CUB | CARS | INET | CUB |
| Baseline Resnet50 | 57.7 | 54.4 | 37.1 | 98.0 | 97.8 | 99.4 | 74.4 | 75.1 | 71.6 | 34.0 |
| glm-saga$_5$ | 55.4 | 51.8 | 35.8 | **97.8** | **97.6** | **99.4** | 74.0 | 74.5 | 71.7 | 2.5 |
| PIP-Net | **99.1** | **99.0** | - | 75.6 | 62.9 | - | **99.5** | **99.5** | - | 6.7 |
| ProtoPool | 24.5 | 30.7 | - | 96.9 | 96.0 | - | 76.7 | 78.9 | - | 13.9 |
| SLDD-Model | 88.2 | 88.6 | 64.7 | 96.2 | 95.6 | 98.6 | 87.2 | 89.7 | **93.4** | 29.2 |
| Q-SENN | 93.3 | 94.4 | **82.0** | 95.5 | 94.8 | 98.7 | 93.0 | 94.2 | 92.6 | 23.4 |
| QPM (Ours) | 90.1 | 89.6 | 64.1 | 97.0 | 96.5 | 99.1 | 96.0 | 97.7 | 89.3 | **47.9** |

scale-dependency (table 8). Therefore, we propose the *Scale-Invariant-Diversity@5* (SID@5)

$$\hat{M}_{i,j}^d = \frac{M_{i,j}^d}{\frac{1}{w_M h_M} \sum |\mathbf{M}^d|} \quad \hat{S}_{i,j}^d = \frac{e^{\hat{M}_{i,j}^d}}{\sum_{m,n} e^{\hat{M}_{m,n}^d}} \tag{15}$$

$$\text{SID@5} = \frac{\sum_{i=1}^{h_M} \sum_{j=1}^{w_M} \max(\hat{S}_{i,j}^1, \hat{S}_{i,j}^2, \ldots, \hat{S}_{i,j}^5)}{5}, \tag{16}$$

where $\hat{\mathbf{S}}^d$ refers to the result of softmax applied to the $d$-th highest weighted feature map $\mathbf{M}^d$, scaled by its absolute mean. A high SID@5 is visible in figs. 1 and 4, as the 5 features used for each class, localize on very different regions in the image.

## 4.3 RESULTS

This section discusses the experimental results. The usual metrics for compactness-based globally interpretable models are shown in table 2. For the fine-grained datasets, QPM is among the most compact models while showing the highest accuracy, thus setting the state of the art for interpretable models. On ImageNet-1K, where prototype-based methods are not even applicable, QPM is only marginally beaten by *Q-SENN*, which uses compute-intensive iterations and negative reasoning for some classes, which significantly hinders interpretability. A runtime analysis is shown in appendix F. The results for the interpretability metrics are shown in table 3. Note that glm-saga$_5$ and PIP-Net are hardly comparable, as glm-saga$_5$ uses the uninterpretable features of a black-box model and PIP-Net learns very localized class-detectors, with some features activating to 99% on just a single class. In contrast, QPM achieves excellent values across all metrics and datasets in this multicriterial task of self-explaining neural networks, summarized in fig. 6. Its interpretable class representations, composed of diverse, general and contrastive features, mirror reality, as measured by Structural Grounding. Note that QPM learns grounded representations as shown in figs. 1 and 4 without any additional supervision and is able to communicate the only differentiating factor it uses. QPM's local behavior then follows its faithful global explanations, which leads to trustworthy classifications and predictable errors when the differentiating factor is not present, as in fig. 8. The appendix contains more visualizations, including a discussion of failure cases in appendix E, a discussion on polysemantic features (appendix H), an extension of Structural Grounding to ImageNet-1K (appendix I) and a discussion of limitations and future work (appendix K).

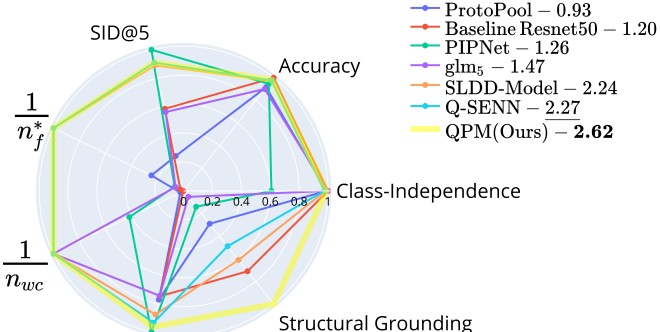

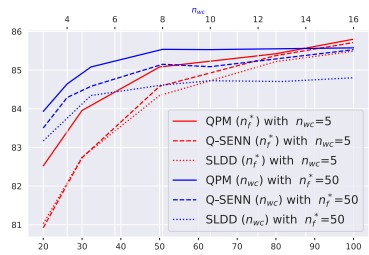

Figure 6: Radar plot across all considered metrics for CUB-2011. Metrics which are preferred to be lower ($n_f^*$, $n_{wc}$) are encoded as $\frac{1}{x}$ and every value is given as a fraction of the maximum. Values in legend are the area of each radar plot.

Figure 7: Compactness-accuracy trade-off compared with *Q-SENN* (dashed) and *SLDD-Model* (dotted) with Resnet50 on CUB-2011. With increasing compactness, QPM's optimal usage of the set $n_f^*$ and $n_{wc}$ becomes more beneficial.

Table 4: Impact of including (✓) additional objectives $Z_B$ (eq. (6)) for locality and $Z_R$ (eq. (5)) to reduce correlation alongside $Z_A$ in eq. (7) on CUB-2011 with Resnet50. Correlation is measured as the average maximally similar feature according to cosine similarity, formulated in appendix C.B.

| $Z_B$ | $Z_R$ | Accuracy ↑ | SID@5 ↑ | Correlation ↓ |
|---|---|---|---|---|
| ✗ | ✗ | 84.6 | 89.0 | 33.9 |
| ✓ | ✗ | 84.4 | **90.3** | 33.5 |
| ✗ | ✓ | 85.0 | 88.5 | **22.7** |
| ✓ | ✓ | **85.1** | 89.6 | 24.6 |

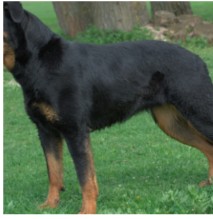

Figure 8: Misclassified example of a QPM, explained in fig. 1: Predictably given the explanation, the model classifies a Doberman as a Rottweiler due to the absent head.

## 4.4 ABLATION STUDIES

This section validates the impact of the individual objectives in the quadratic problem in table 4 and presents the compactness trade-off in fig. 7. We focus on CUB-2011 but observed similar results for other datasets. The compactness-accuracy tradeoff for QPM compared with *Q-SENN* and the *SLDD-Model* is visualized in fig. 7. The global optimization clearly leads to a more effective use of the defined capacity, with the highest uplift in the very high compactness regime, *e.g.* 1.5 percent points at $n_f^* = 20$, where a good selection and assignment naturally has more impact.

The impact of the feature-feature similarity matrix $\mathbf{R}$ and feature selection bias $\mathbf{b}$ is shown in table 4. Incorporating a bias $\mathbf{b}$ for local feature maps further increases the SID@5. On the other hand, reducing feature similarity through $\mathbf{R}$ effectively reduces the correlation between the resulting features, which improves accuracy, as the model uses its capacity more effectively. In summary, the inclusion of the secondary objectives $Z_R$ and $Z_B$ is beneficial for the resulting model, improving the desired aspects not just after solving the QP but also in the resulting model after fine-tuning.

The appendix contains further ablation studies to support our claims, demonstrating the ability to steer (appendix D), validating the choice of correlation as metric for $\mathbf{A}$ (appendix J) and showing the benefits of enforcing exactly $n_{wc}$ features per class (appendix G).

## 5 CONCLUSION

In this paper, we introduced the Quadratic Programming Enhanced Model (QPM). It uses discrete optimization to find an optimal feature selection and assignment of just 5 to each class. With this easy-to-understand assignment, the resulting QPM is more interpretable than previous methods, as it has contrastive faithfully interpretable class-representations, shows Structural Grounding, is steerable, and its features have excellent SID@5, Class-Independence and Contrastiveness. Additionally, it further closes the accuracy gap to the drastically less robust uninterpretable baseline. Figure 6 shows that only QPM excels in all metrics, thus setting a new state of the art for compactness-based interpretable models, while delivering unprecedented global interpretability even to ImageNet-1K.

ACKNOWLEDGMENTS

This work was supported by the Federal Ministry of Education and Research (BMBF), Germany, under the AI service center KISSKI (grant no. 01IS22093C), the Deutsche Forschungsgemeinschaft (DFG) under Germany's Excellence Strategy within the Cluster of Excellence PhoenixD (EXC2122), the European Union under grant agreement no. 101136006 – XTREME. The work has been done in collaboration and partially funded by the Intel Corporation. This work was partially supported by the German Federal Ministry of the Environment, Nature Conservation, Nuclear Safety and Consumer Protection (GreenAutoML4FAS project no. 67KI32007A).

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

Table 5: Statistical overview of datasets. TravelingBirds is used exclusively in the appendix.

| Dataset | CUB-2011 | Stanford Cars | TravelingBirds | ImageNet-1K |
|---------|----------|---------------|----------------|-------------|
| # Classes $n_c$ | 200 | 196 | 200 | 1000 |
| # Training | 5 994 | 8 144 | 5 994 | 1 281 167 |
| # Testing | 5 774 | 8 041 | 5 774 | 50 000 |

## A  APPENDIX

This appendix contains additional details of the implementation details, more results with standard deviations, further experiments on steerability, a discussion of failure cases and the formulation of the Feature Diversity Loss $\mathcal{L}_{\mathrm{div}}$. Further, the quadratic problem is presented in its standard form and the optimality of the found solutions is discussed.

## B  TRAVELINGBIRDS

We use TravelingBirds as an additional dataset to validate our method and for our steering experiments in appendix D. It is based on CUB-2011 and designed to allow the measurement of the robustness of spurious correlations. Specifically, the background of every class in the training set is replaced with an image of a constant class of Places365 (Zhou et al., 2017). For the test set, backgrounds of random classes are used, thus measuring if the model learned to rely on the spuriously correlated background.

## C  IMPLEMENTATION DETAILS

We first describe the implementation on the fine-grained datasets CUB-2011, TravelingBirds and Stanford Cars. All deviating details for ImageNet-1K are included in appendix C.A. The implementation details are similar to the *SLDD-Model* (Norrenbrock et al., 2022), but use the default data for prototype-based methods. Specifically, we use the exact same dense model for both models in our experiments and only alter the following parts of the pipeline with the same hyperparameters for fine-tuning. Therefore, the improved metrics can be attributed to the superior selection and assignment.

**Architectures**  We implement our method using PyTorch (Paszke et al., 2019) and its ImageNet-1K pretrained models as feature extractors. For Resnet50 and Resnet34 we follow *PIP-Net* and use a smaller stride size of 1 for the two last blocks.

**Data**  For training with CUB-2011 and TravelingBirds, the images are first cropped to the ground truth segmentation, following prototype-based methods Nauta et al. (2023); Rymarczyk et al. (2022). After cropping, they are resized to $224 \times 224$ ($299 \times 299$ for Inception-v3). For Stanford Cars and our steerability experiments in table 11, a random crop after resizing one side to the target image size is used instead. Then normalization, random horizontal flip, jitter and *TrivialAugment* Müller & Hutter (2021) is applied. At test time, no augmentation is used and only cropping, random crop replaced by center crop, resizing and normalization is maintained.

**Dense Training**  We fine-tune the pretrained models on the fine-grained datasets using stochastic gradient descent with a batch size of 16 for 150 epochs. The learning rate starts at $5 \cdot 10^{-3}$ for the pretrained layers and $0.01$ for the final linear layer and gets multiplied by $0.4$ every 30 epochs. We set momentum to $0.9$, $\ell_2$-regularization to $5 \cdot 10^{-4}$ and apply dropout with rate $0.2$ to the features. The weighting $\beta$, included in eq. (33), of the Feature Diversity Loss Norrenbrock et al. (2022) is set to $0.196$ for the Resnets, $0.049$ for Inception-v3 and $0.0245$, the highest value we tried for which all dense models converged, for Swin Transformers. Note that the values are scaled with the number of patches in the feature maps, leading to numerical values that do not align conveniently with powers of 10.

**Fine-tuning**  After solving the quadratic problem, the model is trained with the final layer fixed to the sparse assignment of selected features $\boldsymbol{W}^*$ for 40 epochs. The learning rate starts at 100 times the

final learning rate of the dense training and decreases by $60\%$ every 10 epochs. During fine-tuning, momentum is increased to $0.95$ and dropout on the features reduced to $10\%$. For Swin Transformers, the batch size is set to 8 and Layer normalization Ba et al. (2016) is turned off after the dense training has finished, ensuring more unrelated features. All other parameters equal the dense setting.

As the feature maps are the result of ReLU Nair & Hinton (2010), one might expect its values to be strictly $\geq 0$. However, just like for the *SLDD-Model*, the features of QPM are normalized with a fixed mean and standard deviation before fine-tuning begins, resulting in the sub-zero $\min(\boldsymbol{f}_{:,i})$.

**Reproducibility**   For reproducibility, all our experiments with 5 seeds use the integers 16 to 20, ending at 18 for the 3 ImageNet-1K runs, as seed for all random processes.

**Scaling the Objective**   To keep a similar relative weighting across changing $n_{\mathrm{wc}}$ and $n_c$, we also scale the main objective for the quadratic problem $Z_A$ with them

$$Z_A^* = \frac{1000 \cdot Z_A}{n_{\mathrm{wc}} \cdot n_c},\tag{17}$$

ensuring no additional scaling for $n_c = 200$ and $n_{\mathrm{wc}} = 5$.

**Choice of Pretrained Weights**   We use the pretrained Resnet50 weights *V1* of PyTorch for our experiments, as the default *V2* has very class-specific features already, with a Class-Independence of $92.6\%$. For *V2*, a sparse model computed by *glm-saga* (Wong et al., 2021) with just $1.1$ features per class can already achieve $66\%$ accuracy on ImageNet-1K, demonstrating the class-specificness of its features. For Resnet34 and Inception-v3, we use the only available set of weights from PyTorch. For Swin Transformers, we used the original provided weights of PyTorch, as they are suitable for the used image resolution.

## C.A   IMAGENET-1K

Due to computational constraints, we follow the *SLDD-Model*, skip the dense training on ImageNet-1K and directly use the pretrained model as dense model. To facilitate the comparability of metrics between the dense model and our experiments, we use the default strides. For augmentation, we use Lighting noise and omit *TrivialAugment*. Finally, the learning rate of the fine-tuning starts at $\frac{1}{100}$ of the value used for the fine-grained datasets to account for the increased size of the dataset.

## C.B   CORRELATION METRIC

For measuring the effect of reducing correlation between selected features in table 4, the *Correlation* is used:

$$\mathrm{Correlation} = \frac{1}{n_f^*} \sum_{d=1}^{n_f^*} \max_{d \neq d'} \frac{\boldsymbol{f}_{:,d}^T \boldsymbol{f}_{:,d'}}{|\boldsymbol{f}_{:,d}||\boldsymbol{f}_{:,d'}|}\tag{18}$$

## C.C   QUADRATIC PROBLEM

This section presents further details on the quadratic problem and the start solution $\boldsymbol{W}^{\mathrm{Start}}$ for the next iteration of solving the quadratic problem with updated constraints. The start solution is a good, usually optimal, feasible solution for the currently selected set of features $\Lambda$. To simplify the initial iterations, only eq. (9) is considered. The deduplication of eq. (10) is only included after a solution is found that satisfies eq. (9). The start solution is constructed from $\boldsymbol{W}^{n_{\mathrm{wc}}}$ which contains $n_{\mathrm{wc}}$ assignments for each class to the most similar features in $\boldsymbol{A}_{:,\Lambda}$. If the equal distribution of assignments per class is still exclusively optimized for, $\boldsymbol{W}^{\mathrm{Start}} = \boldsymbol{W}^{n_{\mathrm{wc}}}$ is already the start solution. Else, we take care of all classes with equal assignment $C_{\mathrm{equal}}$ in $\boldsymbol{W}^{n_{\mathrm{wc}}}$. Specifically, we remove all

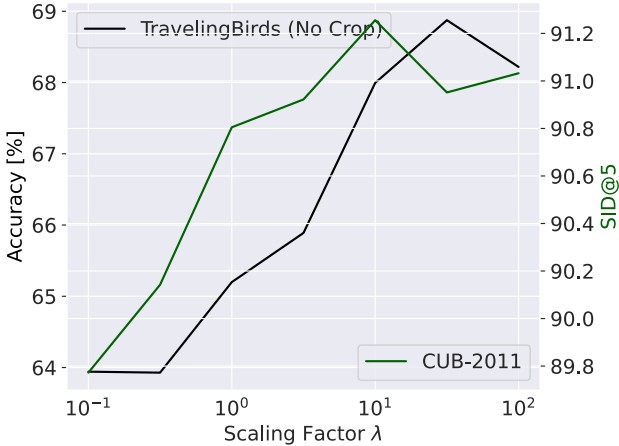

Figure 9: Steerability of the proposed QPM: When increasing the weighting of the bias $\lambda$, the desired metrics, accuracy or SID@5 improve.

duplicate pairs $(c, c') \in C_{\text{equal}}$:

$$
w_{c,d}^{\text{Deduplication}} = \begin{cases} 1 & \text{if} \quad \begin{array}{l} (c, c') \in C_{\text{equal}} \ \& \\ (c, d) = \text{Maxi}(c, c'), \end{array} \\ -1 & \text{if} \quad \begin{array}{l} (c, c') \in C_{\text{equal}} \ \& \\ (c, d) = \text{Mini}(c, c'), \end{array} \\ 0 & \text{else} \end{cases} \tag{19}
$$

$$
\boldsymbol{W}^{\text{Start}} = \boldsymbol{W}^{n_{\text{wc}}} + \boldsymbol{W}^{\text{Deduplication}} \tag{20}
$$

Here, $\text{Mini}(c, c')$ returns the indices to remove, that is of the current assignment with lowest similarity:

$$
\mathbf{h}^c = (\mathbf{a}_c \circ \boldsymbol{w}_c^{n_{\text{wc}}}) \circ \boldsymbol{s} + \max(\mathbf{a}_c \circ \boldsymbol{w}_c^{n_{\text{wc}}}) \cdot (\mathbf{1} - \boldsymbol{s}) \tag{21}
$$

$$
\text{Mini}(c, c') = \begin{cases} c, \text{argmin}(\mathbf{h}^c) & \text{if} \quad \min(\mathbf{h}^c) \leq \min(\mathbf{h}^{c'}) \\ c', \text{argmin}(\mathbf{h}^{c'}) & \text{else} \end{cases} \tag{22}
$$

Here, $\boldsymbol{s}$ is the selection vector and ensures that all changes only apply to the selected features. Similarly, $\text{Maxi}(c, c')$ returns the indices of the assignment to add, which has the highest similarity of the the not currently assigned features:

$$
\mathbf{h}^c = (\mathbf{a}_c \circ \mathbf{cand}(c)) \circ \boldsymbol{s} \tag{23}
$$

$$
\text{Maxi}(c, c') = \begin{cases} c, \text{argmax}(\mathbf{h}^c) & \text{if} \quad \max(\mathbf{h}^c \geq \max(\mathbf{h}^{c'}) \\ c', \text{argmax}(\mathbf{h}^{c'}) & \text{else} \end{cases} \tag{24}
$$

The candidate function

$$
\mathbf{cand}(c) = (\mathbf{1} - \boldsymbol{w}_c^{n_{\text{wc}}}) \cdot \mathbf{wbnd}(c, \boldsymbol{w}^{n_{\text{wc}}}) \tag{25}
$$

checks that the assignment is not made yet and the would-be-no-duplicate function $\text{wbnd}(c, \boldsymbol{W}^{n_{\text{wc}}})_d \in \{0, 1\}$ further ensures that the addition of the assignment of class $c$ to feature $d$ would introduce no duplicate, returning 0 in that case. While this technically does not guarantee an optimal solution, first only finding the solution with $n_{\text{wc}}$ assignments per class and then deduplicating ensures that the number of duplicates is quite low already, which usually leads to finding the optimal feasible start solution.

## D  STEERABILITY

This section is concerned with the ability of the practitioner to steer the model towards desired biases using the feature bias $\boldsymbol{b}$. For example, if a human recognizes the erroneous focus on the background

of a trained QPM, enabled through global interpretability, the feature bias $\boldsymbol{b}^{\text{Center}}$ (eq. (26)) can be used to steer the model towards more centered features.

$$b_d^{\text{Center}} = -\frac{1}{n_T \sum_j f_{j,d}} \sum_{j=1}^{n_T} \frac{1}{1 + d_e(\boldsymbol{M}_d^j)} f_{j,d} \qquad (26)$$

where $d_e$ computes the distance between the maximum of the $j$-th sample's map $\boldsymbol{M}_d^j$ at $(x, y)$ and the closest edge:

$$d_e(\mathbf{M}_d^j) = \min(|x - w_M|, x - 1, |y - h_M|, y - 1) \qquad (27)$$

The resulting improved accuracy on TravelingBirds with $\lambda = 10^{\frac{3}{2}}$, shown in table 11, demonstrates this steerability. Setting $\lambda$ allows a precise weighting of the emphasis put on the bias. This direct control for both the center and diversity bias is visualized in fig. 9 and allows the incorporation of any feature-level bias $\boldsymbol{b}$.

## E  FAILURE CASES

This section presents examples where QPM predicts wrongly. For that, fig. 10 shows exemplary images of Rottweiler and Doberman with classification results of the probed QPM trained on ImageNet-1K and with global explanations in figs. 1 and 21 to 23. Note that the accuracy across the two classes is $87\%$, well above the average, reflected in correct classifications across poses, backgrounds and settings in figs. 10a and 10b. Additionally, fig. 11 shows the GradCAM (Selvaraju et al., 2020) visualizations and demonstrates that QPM always focuses on the dog in the image. For the erroneous predictions, the model behaves just like the global explanations would indicate. Rottweiler and Doberman may be swapped, if the head is occluded as in figs. 10c and 10g or in a difficult pose to gauge the shape, shown in figs. 10d and 10h. Since the Black and tan coon hound is assigned both head features of Rottweiler and Doberman, they can also be confused when primarily the head is visible, demonstrated in figs. 10e and 10i. Finally, figs. 10f and 10j seem to contain one of the many (Northcutt et al., 2021) wrongly labeled samples in ImageNet-1K. QPM also robustly classifies wrongly labeled data, as the global explanation would suggest. Figures 12 and 13 show the feature activations of Greater Swiss Mountain Dog and Rottweiler on fig. 10f and other class examples, further suggesting that it is indeed a typical Greater Swiss Mountain rather a Rottweiler for the probed QPM, as the features of the former localize on the expected regions, whereas most Rottweiler features barely activate. Finally, fig. 14 shows further test examples for the model explained in fig. 4 and demonstrates that the model does not predict Bronzed Cowbird if the differentiating red eye is not present in the image. In summary, QPM's local behavior robustly follows the faithful global explanations, which can lead to predictable faulty classifications in case of occlusion or difficult pose.

## F  RUNTIME ANALYSIS

This section discusses the time it takes to obtain a QPM, compares it to competing models and discusses the impact of $n_f^*$ on it. Figure 15 demonstrates that the optimization time strongly increases when increasing $n_f^*$. However, for the probed datasets, going beyond 50 features seems not to be necessary, as the accuracy only improves negligibly, while the interpretability is harmed: Features become less general and there will be fewer class representations with high overlap, which allow for the most intuitive interpretation. One can further optimize this using suitable priors, which we do not include in this work, as the interpretability and additional accuracy decreases with increasing $n_f^*$. It is however an avenue for future work, when datasets with sufficient complexity are published. Table 6 compares the time to obtain the interpretable model between QPM, *Q-SENN* and *SLDD-Model*. *Q-SENN* and *SLDD-Model* start with a feature selection, that takes 15 minutes on CUB-2011 and roughly 500 minutes on ImageNet-1K. They both use *glm-saga* for feature selection and computing the sparse matrix and are thus scaling with number of samples $n_T$, which QPM is invariant to, as that dimension is summarized in the constants.

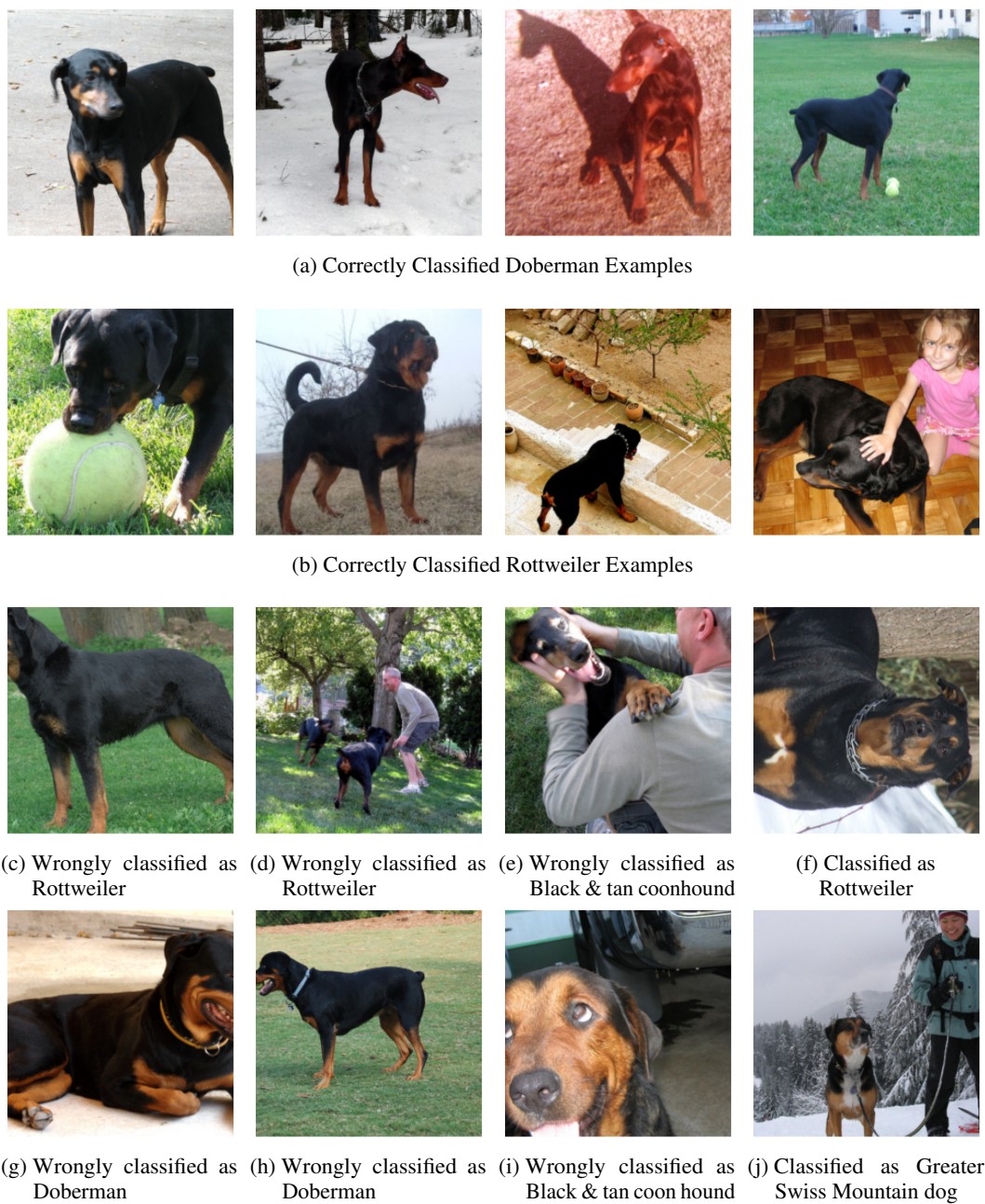

(a) Correctly Classified Doberman Examples

(b) Correctly Classified Rottweiler Examples

(c) Wrongly classified as Rottweiler  (d) Wrongly classified as Rottweiler  (e) Wrongly classified as Black & tan coonhound  (f) Classified as Rottweiler

(g) Wrongly classified as Doberman  (h) Wrongly classified as Doberman  (i) Wrongly classified as Black & tan coon hound  (j) Classified as Greater Swiss Mountain dog

Figure 10: Examples for correctly and wrongly (according to ground truth labels) classified examples of the QPM with global explanations shown in figs. 1 and 21 to 23. Figures 10b to 10f (rows 1 and 3) show Doberman labeled images. Figures 10a and 10g to 10j (rows 2 and 4) display Rottweiler labeled images. The resulting classifications match the expected behavior based on the global explanations. As the explained QPM uses the head to differentiate between Doberman and Rottweiler (fig. 1), they can be confused when it is occluded (figs. 10c and 10e) or in a difficult pose (figs. 10d and 10h). As the black and tan coon hound is assigned the same head features (fig. 23), they get confused, if only the head is visible (figs. 10e and 10i). Finally, the probed QPM correctly classifies according to its explanations (figs. 10a and 10b), also on wrongly labeled samples (figs. 10f and 10j).

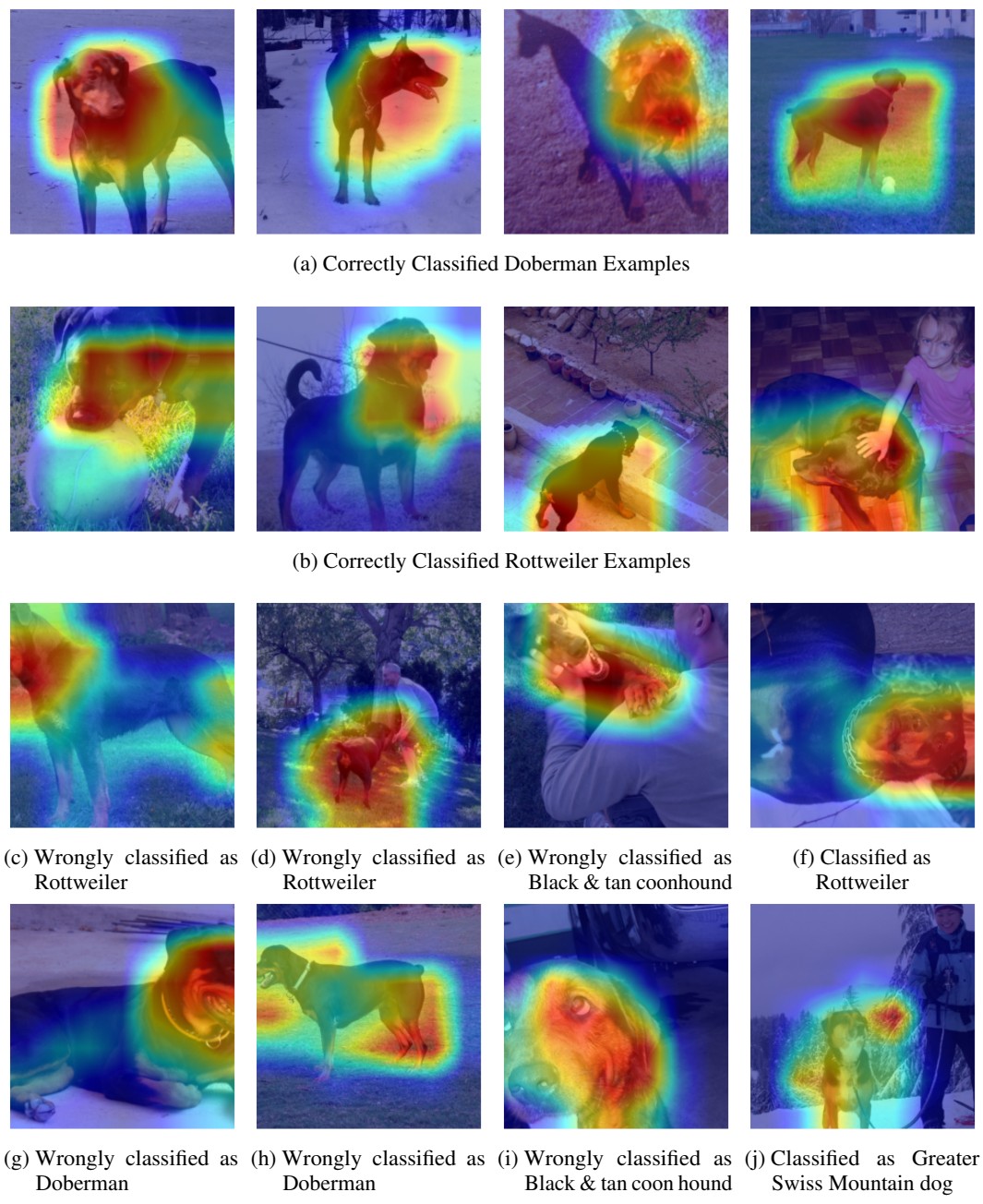

(a) Correctly Classified Doberman Examples

(b) Correctly Classified Rottweiler Examples

(c) Wrongly classified as Rottweiler
(d) Wrongly classified as Rottweiler
(e) Wrongly classified as Black & tan coonhound
(f) Classified as Rottweiler

(g) Wrongly classified as Doberman
(h) Wrongly classified as Doberman
(i) Wrongly classified as Black & tan coon hound
(j) Classified as Greater Swiss Mountain dog

Figure 11: Gradcam Visuliazations for fig. 10.

## G   IMPACT OF EVEN SPARSITY

This section discusses the impact of enforcing exactly $n_{\mathrm{wc}}$ features per class, rather than on average. For that, we trained a model without this constraint, but instead with $\mathbf{1}^T \boldsymbol{W} \boldsymbol{s} = n_{\mathrm{wc}} n_c$ enforcing an average sparsity. To counteract the uneven number of features per class, every class got a bias, that is linear to the number of features it is below the average. In prior experiments, various forms of counteracting the uneven assignment with a bias have performed similarly. Table 7 shows that the even assignment is beneficial for the accuracy. Further, the even assignment boosts interpretability as it leads to more classes that can be contrasted easily and does not introduce an unintuitive bias term. Additionally, fig. 16 demonstrates that classes, which are assigned to fewer features, cause these

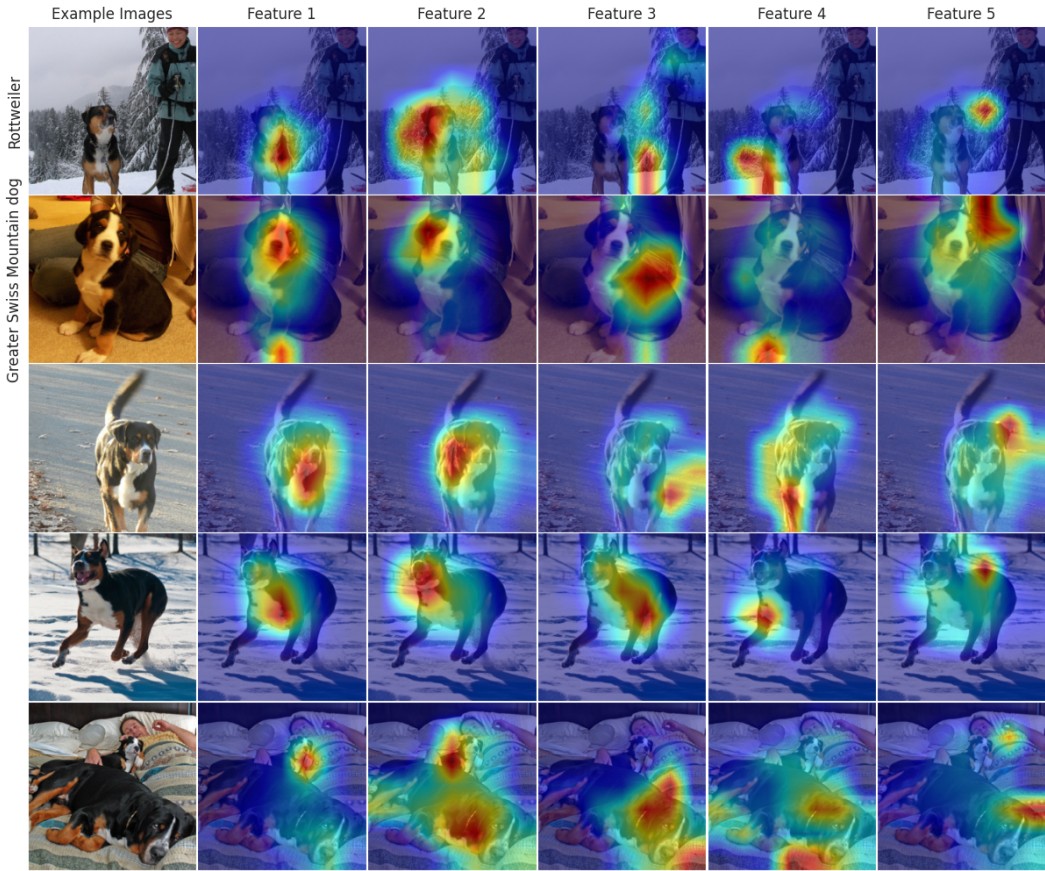

Figure 12: Features of Greater Swiss Mountain Dog and their localization on the sample in fig. 10j (first row), that is presumably falsely labeled Rottweiler. The 4 lower rows contain examples of Greater Swiss Mountain Dog and the features consistently localize around semantically similar regions, also on the Rottweiler labeled one.

Table 6: Time in minutes between finishing dense training and obtaining the final model. First value is time for optimization, second time spent fine-tuning. Following *Q-SENN*, we use the *fast* setting for ImageNet-1K. *Q-SENN* trains for 70 epochs, instead of 40, in total during fine-tuning and does 4 iterations of *glm-saga*. Note that every method runs exclusively on a GPU server, except for the QP optimization, which can be done on just a CPU.

| Method | CUB | INET |
|---|---|---|
| SLDD-Model | $(15 + 22) + 78 = \mathbf{115}$ | $(500 + 3000) + 3600 = 7100$ |
| Q-SENN | $(15 + 4 * 22) + 78 * 7/4 \approx 240$ | $(500 + 4 * 100) + 7/4 * 3600 = 7200$ |
| QPM (Ours) | $210 + 78 = 298$ | $660 + 3600 = \mathbf{4260}$ |

features to become less general for QPM and Q-SENN, which hurts interpretability and potentially accuracy. Figure 17 also visualizes that Q-SENN always learns to represent classes with a huge variety in the number of assigned features, necessarily leading to hardly interpretable representations. Nevertheless, the impact is disparate on the two datasets and the accuracy increase is not significant on Stanford Cars. Future work might investigate if datasets with classes of varying complexity will benefit from representing classes with a suitable number of features and how this can be combined with contrastive globally interpretable class representations.

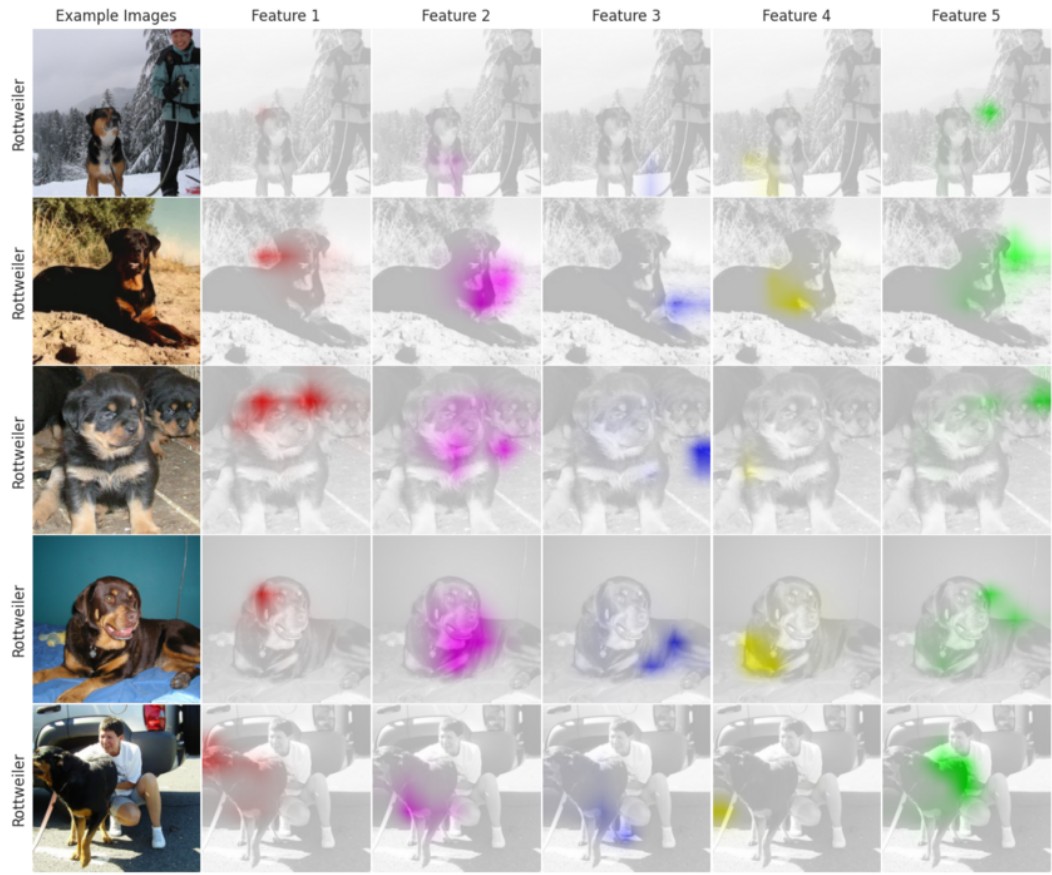

Figure 13: Features of Rottweiler and their localization, scaled by column, on the sample in fig. 10j (first row), that is presumably falsely labeled Rottweiler. The 4 lower rows contain examples of Rottweiler. The color code is consistent with fig. 1 and features 4 and 5 are shared with Greater Swiss Mountain Dog.

Table 7: Accuracy with or without exactly $n_{\mathrm{wc}}$ features per class (eq. (3)). Instead, on average $n_{\mathrm{wc}}$ features per class are used.

| Method | CUB | CARS |
|---|---|---|
| without eq. (3) | 84.3±0.2 | 91.6±0.3 |
| QPM (Ours) | **85.1**±0.3 | **91.8**±0.1 |

## H  POLYSEMANTIC FEATURES

This section discusses the phenomenon of polysemantic features and how it relates to QPM. Like all deep learning models (Scherlis et al., 2022) not specifically designed to prevent polysemanticity, QPM learns polysemantic features. It refers to individual neurons activating on not just one concept $c$ but rather on $n$ seemingly unrelated ones. While it is an active area of research, their emergence can likely be attributed to being an effective solution to the training objective. On many training samples, the impact on the loss can be fairly low, if a polysemantic feature activates on any of its $n$ meanings. The only exception occurs, when it activates on samples, where its activation contributes significantly to a class that is already showing a lot of activation. While this is typically very difficult to analyze, the interpretable structure of QPM can offer more insights, as it enables a reliable metric on which to gauge how strongly the activation on another concept would affect the loss: The similarity in QPM's class representation space. Our hypothesis is that QPM learns features that are locally monosemantic, while being globally polysemantic. Around a class, e.g., Bronzed Cowbird, we expect the features

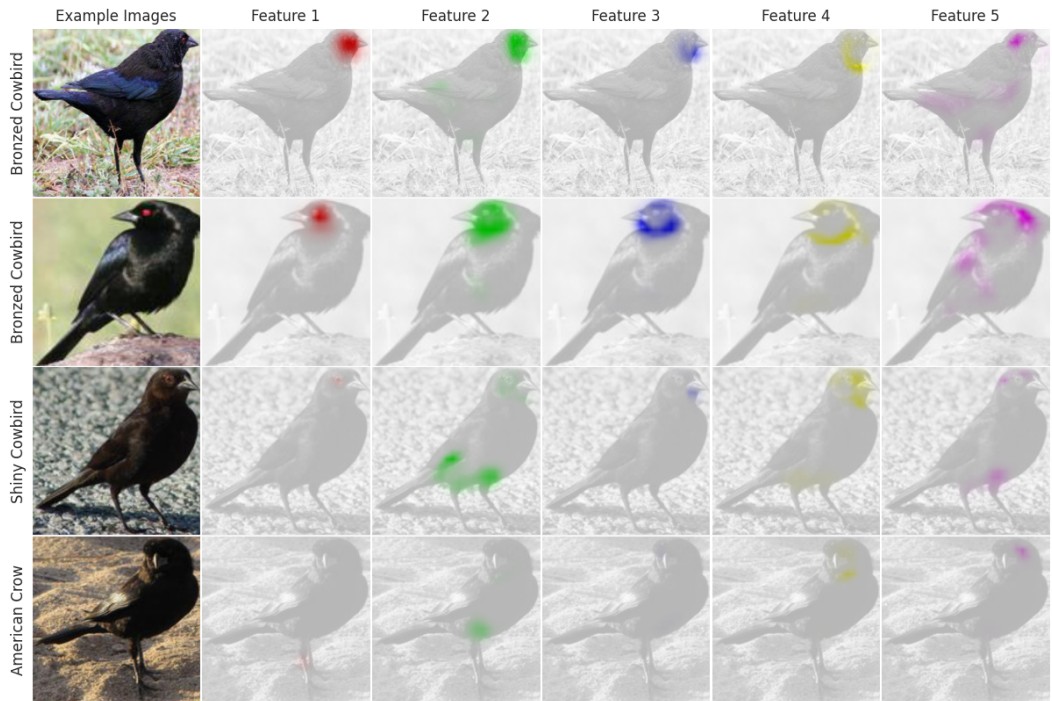

Figure 14: Features of Bronzed Cowbird, explained and compared with Shiny Cowbird in fig. 4, the predictions of the QPM, and their localizations, normed across column, on Bronzed Cowbird labeled test samples. When all features, including the red eye (feature 1) are visible (rows 1 and 2), the model is correct. However, as expected from the global explanation, without the red eye it can be wrong and confuse e.g. Shiny Cowbird with it. The probed QPM represents American Crow with features 2,4,5 and 2 further not shown features, that localize on wing and beak of crows.

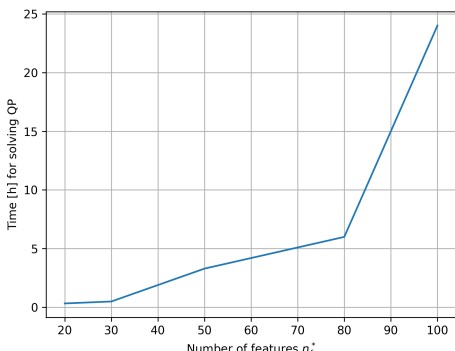

Figure 15: Time it takes to optimize QP for models with varying $n_f^*$ in fig. 7.

to only activate on one of the $n$ concepts that they activate on across the entire dataset. As this is generally fairly difficult to measure, we show anecdotal evidence for this in fig. 18. It shows the Feature Alignment metric from *Q-SENN* (Norrenbrock et al., 2024) relative to the similarity to the Bronzed Cowbird, measured as the number of its features that classes do not share. Specifically, given the training features $\boldsymbol{F} \in \mathbb{R}^{n_T \times n_f}$ s,

$$A_{a,j}^{\text{gt}} = \frac{1}{|\rho_{a+}|} \sum_{i \in \rho_{a+}} F_{i,j}^{\text{train}} - \frac{1}{|\rho_{a-}|} \sum_{i \in \rho_{a-}} F_{i,j}^{\text{train}} \tag{28}$$

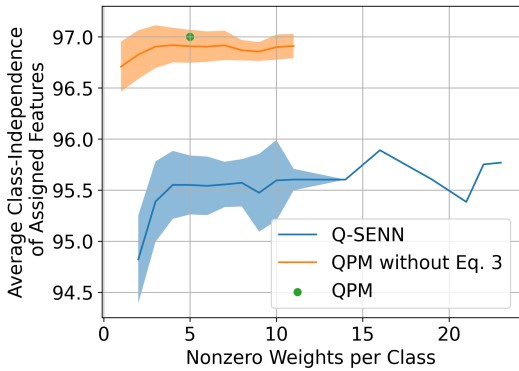

Figure 16: Average Class-Independence of Assigned Features on CUB-2011 as function of the number of features assigned to the class. The distribution of the sparsity is shown in fig. 17.

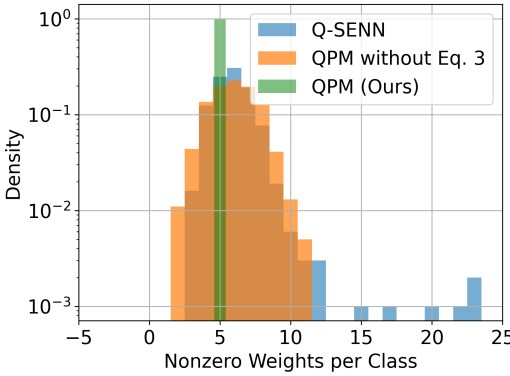

Figure 17: Distribution of Nonzero Weights per Class on CUB-2011. For each method, all $1000$ classes from $5$ seeds are shown.

describes the average difference in activations when an annotated attribute is present, encoded in $\rho_{a+}$ or $\rho_{a-}$ for absent. Norrenbrock et al. (2024) then scales the difference by the average zero based activation and reports the average maximum per feature:

$$r = \frac{1}{n_f^*} \sum_{j=1}^{n_f^*} \frac{n_T}{\sum_{l=1}^{n_T} F_{l,j}^{\text{train}} - \min_l F_{l,j}^{\text{train}}} \max_i A_{i,j}^{\text{gt}}. \tag{29}$$

For our analysis, we limit these formulas to just the attribute red eye color $red - eye$ and only consider the one feature $k$ detecting it for Bronzed Cowbird:

$$r_{red-eye}(x) = \frac{n_T}{\sum_{l=1}^{n_T} F_{l,k}^{\text{train}} - \min_l F_{l,k}^{\text{train}}} A_{red-eye,k}^{\text{gt}}(x). \tag{30}$$

The x-axis additionally describes a filtering applied to the features and attributes based on the similarity of the label, where a sample is considered for computing $A_{red-eye,k}^{\text{gt}}(x)$ if the annotated label shares at least $5 - x$ features with Bronzed Cowbird. Figure 18 demonstrates that the feature clearly detecting the red eye of the Bronzed Cowbird is indeed quite sensitive to its presence when the ground truth label is similar to the class, while it globally loses that sensitivity as it also detects other concepts of classes further away from Bronzed Cowbird.

## I    STRUCTURAL GROUNDING ON IMAGENET

This section is concerned with evaluating a metric similar to Structural Grounding on ImageNet-1K. It is based on comparing the class similarities in reality $\boldsymbol{\Psi}^{gt}$ with the $\boldsymbol{\Psi}^{Model}$ ones learned by our

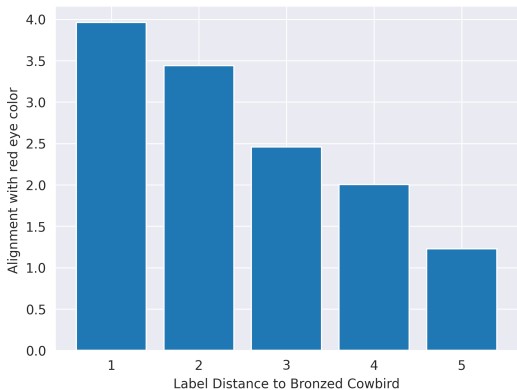

Figure 18: Feature Alignment metric from Q-SENN Norrenbrock et al. (2024) for the red-eye feature (marked in red) in fig. 4 and the attribute red eye color $r_{red-eye}(x)$. The x-axis describes to which samples the computation is limited, e.g. $x = 2$ describes computing the metric only on samples whose label is represented using up to 2 other features. On the right, $x = 5$ refers to the usual global feature alignment. The probed QPM learned a polysemantic, but locally monosemantic feature. When differentiating between bronzed and shiny cowbird only ($x = 1$), the feature value increased by almost 4 times its mean, if the attribute is annotated to be present.

model. Structural Grounding relies on the annotations of CUB-2011 to compute $\mathbf{\Psi}^{gt}$. However, there are no such annotations on a fine-grained scale for ImageNet-1K. Therefore, we use the similarity of the text-names in CLIP (Radford et al., 2021) as proxy to obtain our ground-truth class similarities. Specifically, we compute the cosine similarity $\mathbf{\Psi}^{clip} \in [-1,1]^{n_c \times n_c}$ between the text embeddings of the class names, obtained from the powerful pretrained *ViT-L-14*, that is broadly used, e.g. to condition Stable Diffusion XLPodell et al. (2023). We always take the first description given for every class.

When inspecting the most similar classes, several issues are apparent. Many of them include shared tokens or words, e.g., *ski* and *ski mask*, *lion* and *sea lion*, *rule* and *stole* or *digital clock* and *wall clock*. While some of these indeed describe a similar class, e.g., *giant Schnauzer* and *Standard Schnauzer*, others do not. Including classes without high similarity as ground-truth similar classes harms the quality of the evaluation drastically and demonstrates the value of having human annotations. Another issue in the clip similarities is that fine-grained knowledge about the classes seems to be less dominant than literal exact matching, as with higher similarity only more commonly used terms are correctly associated with similar terms, e.g., *Orangutan* and *Gorilla*, but not *Rottweiler* and *Doberman*. Therefore, the number of similar classes to consider is set to 1250, as the latter pair ranks at position 828 and we definitely consider it as a similarity worth measuring. Notably, this pair is ranked behind pairs such as *hog* and *tank*, *lemon* and *yawl* or *hamster* and *snail*, further demonstrating the weakness of the language model to exactly model the similarities. The final apparent issue lies in the ambiguity of class names which leads to *crane* appearing twice as class name, once referring to birds, once to a machine on a construction site. Notably, the distribution of class sparsity has significant impact. While QPM is limited to a class similarity of up to $80\%$, due to deviating in at least one feature with all features sharing the same weight and every class being represented by $n_{wc} = 5$ features, *SLDD-Model*, *Q-SENN* and glm-saga₅ all exhibit multiple (14, 9 and 5) class pairs, that have a class similarity of above $99\%$. The *SLDD-Model* for instance repeatedly represents classes with one feature with positive weight and one with an extremely low negative weight, resulting even in cosine similarities of 1 due to floating point precision. While this generally hurts interpretability, it can be beneficial for Structural Grounding.

Despite these issues, Table 9 shows that QPM still performs comparatively to *SLDD-Model* and *Q-SENN* with their extremely high similarities and learns significantly more aligned representations than the dense baseline, even on ImageNet-1K. Future work might incorporate a more fine-grained class hierarchy, building upon the very general WordNet, into this metric or profit off of further improved language models.

Table 8: Results for diversity@5 (Norrenbrock et al., 2022) demonstrating its weakness to capture the locality of the by-design very local features of PIP-Net. Note that 20 is the worst possible value.

| Method | CUB | CARS |
|---|---|---|
| Baseline Resnet50 | $61.1\pm0.4$ | $57.4\pm0.3$ |
| glm-saga$_5$ | $55.3\pm0.5$ | $52.6\pm0.3$ |
| PIP-Net | $20.5\pm0.0$ | $20.5\pm0.0$ |
| ProtoPool | $25.5\pm0.4$ | $23.4\pm0.5$ |
| SLDD-Model | $79.2\pm0.3$ | $81.9\pm0.9$ |
| Q-SENN | $87.0\pm0.5$ | $89.6\pm0.3$ |
| QPM (Ours) | $\mathbf{89.9}\pm0.2$ | $\mathbf{91.4}\pm0.3$ |

Table 9: Structural Grounding based on Clip Similarities on ImageNet

| Method | Structural Grounding |
|---|---|
| Dense Resnet50 | $17.9\pm0.0$ |
| glm-saga$_5$ | $10.3\pm0.0$ |
| *SLDD-Model* | $\mathbf{36.9}\pm0.4$ |
| Q-SENN | $33.2\pm0.2$ |
| QPM (Ours) | $\underline{34.5}\pm0.6$ |

## J  IMPACT OF CLASS-FEATURE SIMILARITY METRIC

This section contains an ablation study on the choice of Pearson correlation as metric for the feature-class similarity matrix $A$. While it captures the desired linear relationship, that is also utilized during the following predictions, an intuitive alternative is the Area under the receiver operating characteristic curve (AUROC), which is highly non-linear and frequently used to capture the predictive power with a varying threshold. Table 10 shows that AUROC is also suitable but inferior to the simple correlation.

## K  LIMITATIONS AND FUTURE WORK

This section discusses limitations for the proposed QPM and avenues for future work.

In this work, QPM is applied to the generally available and typical datasets for image classification, with ImageNet-1K indicating broad applicability. However, QPM's high interpretability is especially beneficial for high-stakes applications such as the medical domain or autonomous driving, where each individual situation can not be accessed by an expert. Rather, after training the QPM and before deploying it to cars, its class explanations can be obtained to gain insights into whether it is right for the right reasons and if these are robust to all deployment conditions. Thus, applying QPM to suitable high-stakes applications is a promising avenue for future work. However, to our knowledge, there is no suitable dataset from these domains published yet.

A limitation of our QPM in its current form lies in its inability to model negative assignments. Compared to the *SLDD-Model* and *Q-SENN*, which use negative weights, it is evident that the varied datasets used in this paper, do not require it. Further, while we believe that it is generally preferable to represent classes only using positive assignments, as e.g., also done by recent prototypical models (Nauta et al., 2023; Rymarczyk et al., 2022), one can think of other datasets where negative reasoning may be superior. If, e.g., all classes in a dataset containing birds had a black beak, except for one with all other colors, it would likely be the most efficient solution to represent that one with a negative assignment on a feature activating on black beaks, rather than have every other class positively assigned to it, which the current QPM might do. Thus, future work may incorporate negative assignments into the optimization, which might lead to even more compact representations.

As discussed in appendix H, the learned features of our QPM are generally polysemantic, while potentially being monosemantic locally. For aligning them with human concepts, all post-hoc

Table 10: Accuracy with different criteria used as Feature-Class Similarity matrix $\boldsymbol{A}$

| $\boldsymbol{A}$ Metric | CUB | CARS |
|---|---|---|
| AUROC | 84.8±0.2 | 91.6±0.2 |
| Correlation (Ours) | **85.1**±0.3 | **91.8**±0.1 |

methods, such as TCAV (Kim et al., 2018), Clip-Dissect (Oikarinen & Weng, 2023), or the alignment methods from *SLDD-Model* or *Q-SENN* can be applied. Notably, aligning a feature with their human concepts is more beneficial for QPM than it is for e.g., black-box models, as they are used in an intuitively interpretable way. Further, the interpretable assignment can even help with alignment, as shown in appendix H. Nevertheless, polysemantic features are a challenge for interpretability and future work in this direction can focus on preventing their emergence while still using them in an interpretable way or robustly measuring alignment to multiple concepts.

For many explanations from our QPM, a saliency map for its individual features is used. While we typically just visualize each individual feature map via upscaling, resulting in a comparable resolution to GradCAM (Selvaraju et al., 2020), other saliency methods, like Integrated Gradients (Sundararajan et al., 2017), LRP (Binder et al., 2016) or RISE (Petsiuk et al., 2018) can be applied. Because QPM is backbone independent, even models with built-in more faithful saliency maps such as B-cos Networks (Böhle et al., 2023) can be used. Since the use of these features is easy-to-interpret, evaluating the localizations of our model should focus on the feature explanations rather than class-level ones. Future work might incorporate these faithful saliency maps to measure insertion or deletion methods, akin to those used for class-level saliency maps (Petsiuk et al., 2018). Ideally, one is able to overcome the issue of moving out-of-distribution with removing pixels (Hooker et al., 2019). Finally, the contrastive nature of QPM's features might lead to an intuitive threshold that can be used during the removal of pixels, similar to how previous metrics try to change the class prediction.

## L   DETAILED RESULTS

This section contains detailed results with standard deviations, including experiments with Resnet34, Inception-v3, Swin-Transformer-small and Swin-Transformer-tiny, in Suppl. table 11 to table 22. The good results across architectures demonstrate an independence between backbone and our proposed method. They further seem robust as the difference in mean is usually large compared to the standard deviation. Further, figs. 21 and 22 show how the features of fig. 1 continue to localize on the same human attribute across different poses. Additionally, we included the activations of these features on images of another class in fig. 23 to showcase the global interpretability enabled through the binary assignment of more interpretable features. Instead of the blue and green feature, this probed QPM recognizes the Black and Tan Coonhound through both doberman-like and rottweiler-like head features, as well as a neck that is also assigned to pandas or bears. Figures 19 and 20 additionally include examples for contrastive class representations learned on Stanford Cars and TravelingBirds. Finally, table 8 contains results for diversity@5, to quantify its inability to capture the high spatial diversity of PIP-Nets class detectors.

## M   FEATURE DIVERSITY LOSS

This section further describes the Feature Diversity Loss $\mathcal{L}_{\text{div}}$, proposed in Norrenbrock et al. (2022). It is defined per sample, for which the model predicted the class $\hat{c} = \arg\max(\boldsymbol{y})$ and ensures a local diversity of the used feature maps $\boldsymbol{M} \in \mathbb{R}^{n_f \times w_M \times h_M}$.

$$\hat{s}_{ij}^d = \frac{\exp(m_{ij}^d)}{\sum_{i'=1}^{h_M}\sum_{j'=1}^{w_M}\exp(m_{i'j'}^d)} \frac{\boldsymbol{f}_d}{\max \boldsymbol{f}} \frac{|w_{\hat{c},d}|}{\|\boldsymbol{w}_{\hat{c}}\|_2} \tag{31}$$

$$\mathcal{L}_{\text{div}} = -\sum_{i=1}^{h_M}\sum_{j=1}^{w_M}\max(\hat{s}_{ij}^1, \hat{s}_{ij}^2, \ldots, \hat{s}_{ij}^{n_f}) \tag{32}$$

Equation 31 employs the softmax function to normalize the entries $m_{ij}^l$ of the feature maps $\boldsymbol{M}$ across spatial dimensions. It then scales the maps to emphasize visible and significant features, maintaining

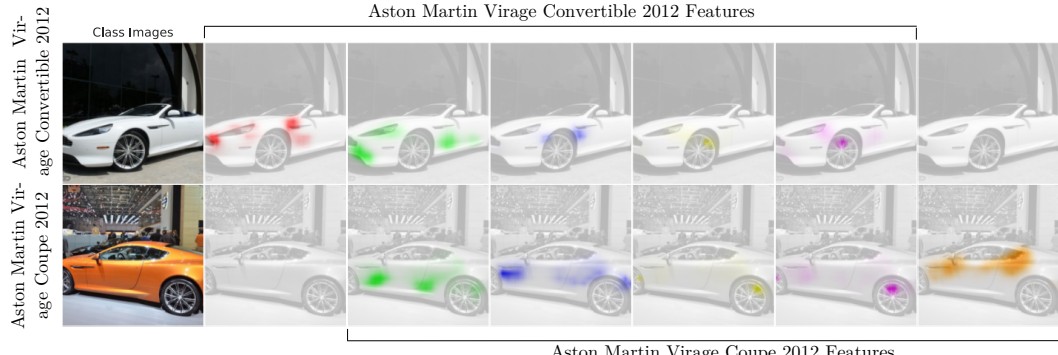

Figure 19: Faithful global interpretability of our QPM trained on Stanford Cars: Without any additional supervision, QPM learns to represent the Convertible and Coupe Variant using 5 diverse and general features.

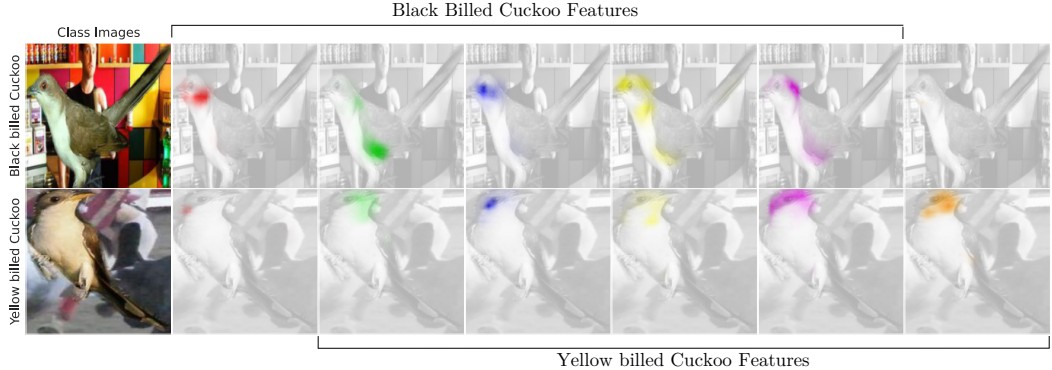

Figure 20: Faithful global interpretability of our QPM trained on TravelingBirds: Without any additional supervision, QPM learns to represent the Yellow and Black billed Cuckoo using 5 diverse and general features, correctly ignoring the correlated background.

the relative mean of $M$ while weighting them according to the predicted class. Equation 32 then applies cross-channel-max-pooling of the normalized and scaled feature maps $\hat{S}$. The result is negatively weighted and thus encourages the model to learn features that localize on different image regions. The resulting total training loss is

$$\mathcal{L}_{\text{final}} = \mathcal{L}_{CE} + \beta\mathcal{L}_{\text{div}}, \tag{33}$$

with $\beta \in \mathbb{R}_+$ as weighting factor.

## N    OPTIMALITY OF SOLUTION

In order to test the optimality of our solution, we try to solve the problem without our relaxation in eq. (9) with more compute and time. We used 3 days and 250 GB on an AMD EPYC 72F3 to solve the problem globally across 5 seeds on CUB-2011 with a target gap to optimality of $1\%$ to ensure sufficient deduplication. The time limit was left to 3 hours for one iteration, as otherwise multiple iterations would not finish. Across the 5 seeds used for QPM, the average obtained objective value for the global problem was $0.5\%$ above the one computed with our simplifications. Similar to our ablations in table 4, the resulting accuracy for the extensively optimized model was not improved, but even $0.1$ percent points lower. As mentioned in section 4.1.1, the objective does not perfectly correlate with downstream metrics, as the constants $A$, $R$ and $b$ only approximate the desired behaviour. However, the average gap to the best bound was still $3.2\%$, with only negligible progress during the final iteration, suggesting that a longer time limit would not significantly improve it. Note, that the best bound might be violating constraints, already added or not. In summary, the

gap between our easy-to-compute solution and an obtainable solution of the global problem is $0.5\%$, which leads to no improved model, with an upper bound on the gap of $3.7\%$ (372 to 386).

## O  STANDARD FORM FOR QUADRATIC PROBLEM

The quadratic problem, described in section 3.1, can be expressed in the standard form for quadratic programming problems. The aim is to optimize quadratic problems of the form $\frac{1}{2}\mathbf{x}^T\mathbf{Q}\mathbf{x} + \mathbf{c}^T\mathbf{x}$ with respect to specified constraints. To describe our quadratic problem in standard form, we therefore define the **variables** $\mathbf{x}$, $\mathbf{Q}$, $\mathbf{c}$ as well as the **constraints**. For notation, $\mathbf{0}^x$ and $\mathbf{1}^x$ describe a vector with $x$ zeros or ones respectively and $\mathbf{0}^{m,n}$ describes a $m \times n$ matrix of zeros.

**Variables**  Let $\mathbf{x}$ be the binary decision variable vector, combining $\boldsymbol{s}$ and the vectorized form of $\boldsymbol{W}$:

$$\mathbf{x} = \begin{bmatrix} \mathbf{s} \\ \text{vec}(\boldsymbol{W}) \end{bmatrix} \in \{0,1\}^{n_f + n_c \cdot n_f}$$

**Objective Function**  The standard objective function includes all objectives:

$$\text{Maximize:} \quad \frac{1}{2}\mathbf{x}^T\mathbf{Q}\mathbf{x} + \mathbf{c}^T\mathbf{x}$$

Here

$$\mathbf{Q} = \begin{bmatrix} -\boldsymbol{R} & \mathbf{0}^{n_f, n_f \cdot n_c} \\ \boldsymbol{A}_{\text{stack}} & \mathbf{0}^{n_f \cdot n_c, n_f \cdot n_c} \end{bmatrix} \tag{34}$$

combines all quadratic objectives and

$$\mathbf{c} = \begin{bmatrix} \boldsymbol{b} \\ \mathbf{0}^{n_f \cdot n_c} \end{bmatrix} \tag{35}$$

the linear term. Here

$$\boldsymbol{A}_{\text{stack}} = \begin{bmatrix} diag(\boldsymbol{a}_1) \\ diag(\boldsymbol{a}_2) \\ \vdots \\ diag(\boldsymbol{a}_{n_c})) \end{bmatrix} \tag{36}$$

connects the vectorized entries of $\boldsymbol{W}$ with $\boldsymbol{A}$.

**Constraints**

1. Constraint for the number of selected features (eq. (2)):

$$\begin{bmatrix} \mathbf{1}^{n_f} \\ \mathbf{0}^{n_f \cdot n_c} \end{bmatrix}^T \mathbf{x} = n_f^* \tag{37}$$

2. No assignments on unselected features:

$$[\textbf{featureSum} \quad \mathbf{0}^{n_f \cdot n_c}](\mathbf{1}^{n_f \cdot (n_c+1)} - \mathbf{x}) = 0 \tag{38}$$

$$\textbf{featureSum} = [\mathbf{0}^{n_f, n_f} \quad \text{FeatureSel}^{n_f}]\,\mathbf{x} \tag{39}$$

where $\textbf{FeatureSel}^{n_f} \in \{0,1\}^{n_f \times n_f \cdot n_c}$ is a matrix of zeros with $\text{FeatureSel}_{i,j} = 1$ where $(j - i) \mod n_f = 0$. The vector $\textbf{featureSum}$ captures the total number of assignments per feature.

3. Constraint for the number of assignments per class (eq. (3)):

$$[\mathbf{0}^{n_c, n_f} \quad \text{UBD}^{n_c, n_f}]\,\mathbf{x} = n_{\text{wc}} \cdot \mathbf{1}^{n_c} \tag{40}$$

Where the upper block diagonal matrix

$$\text{UBD}^{n_c, n_f} = \begin{bmatrix} \mathbf{1}^{n_f} & \mathbf{0}^{n_f} & \cdots & \mathbf{0}^{n_f} \\ \mathbf{0}^{n_f} & \mathbf{1}^{n_f} & \cdots & \mathbf{0}^{n_f} \\ \vdots & \vdots & \vdots & \vdots \\ \mathbf{0}^{n_f} & \mathbf{0}^{n_f} & \cdots & \mathbf{1}^{n_f} \end{bmatrix}^T \tag{41}$$

is a block-diagonal matrix with $n_f$ ones per row, one 1 in each of the $n_f \cdot n_c$ columns and $n_c$ total rows.

Table 11: Accuracy without background removal based on the Ground-Truth with Resnet50. QPM is less susceptible to the spuriously correlated backgrounds with and without Center Bias $b^{\text{Center}}$. NA indicates no convergence.

| Method | CUB | TRAVEL |
|---|---|---|
| Baseline Resnet50 | 84.2±0.3 | 33.8±0.6 |
| glm-saga$_5$ | 75.0±0.9 | 35.6±1.4 |
| PIP-Net | 74.9±0.0 | 59.4±1.0 |
| ProtoPool | 75.0±0.3 | NA |
| SLDD-Model | 82.2±0.1 | 62.6±1.6 |
| Q-SENN | 82.8±0.3 | 67.0±0.5 |
| QPM (Ours) | **82.9**±0.1 | 64.7±0.7 |
| w/ Center Bias $b^{\text{Center}}$ | 82.4±0.3 | **68.9**±0.5 |

4. No duplicated classes (eq. (4)):

$$\begin{bmatrix} Eq^{c,c'} & \mathbf{0}^{n_f \cdot n_c} \end{bmatrix} \mathbf{1}^{n_f \cdot (n_c+1)} > 0 \quad \forall c \neq c' \in \{1, \ldots, n_c\} \tag{42}$$

$$Eq_d^{c,c'} = |x_{c \cdot n_f + d} - x_{c' \cdot n_f + d}| \quad \forall d \in \{1, \ldots, n_f\} \tag{43}$$

Table 12: Ablation Studies investigating the impact of incorporating feature-feature similarity through $R$ and locality bias $b$ on CUB-2011 with Resnet50.

| $b$ | $R$ | Accuracy ↑ | SID@5 ↑ | Correlation ↓ |
|---|---|---|---|---|
| ✗ | ✗ | 84.6±0.4 | 89.5±0.2 | 33.9±0.8 |
| ✓ | ✗ | 84.4±0.2 | **90.4**±0.3 | 33.5±9.4 |
| ✗ | ✓ | 85.0±0.3 | 89.4±0.3 | **22.7**±1.1 |
| ✓ | ✓ | **85.1**±0.3 | 90.1±0.3 | 24.6±1.1 |

Table 13: Comparison on compactness and accuracy with Resnet50: QPM shows increased accuracy and compactness. The compactness-accuracy trade-off is shown in fig. 7. Among more interpretable models, the best result is marked in bold, second best underlined.

| Method | Accuracy↑ | | | | Total Features↓ | | | | Features / Class↓ | | | |
|---|---|---|---|---|---|---|---|---|---|---|---|---|
| | CUB | CARS | TRAVEL | IMGNET | CUB | CARS | TRAVEL | IMGNET | CUB | CARS | TRAVEL | IMGNET |
| Baseline Resnet50 | 86.6±0.2 | 92.1±0.1 | 59.5±0.9 | 76.1 | 2048 | 2048 | 2048 | 2048 | 2048 | 2048 | 2048 | 2048 |
| glm-saga$_5$ | 78.0±0.4 | 86.8±0.6 | 58.0±1.2 | 58.0±0.0 | 809±8 | 807±10 | 781±6 | 1627±1 | **5** | **5** | **5** | **5** |
| PIP-Net | 82.0±0.3 | 86.5±0.3 | 74.0±0.4 | - | 731±19 | 669±13 | 744±15 | - | 12 | 11 | 6 | - |
| ProtoPool | 79.4±0.4 | 87.5±0.2 | 38.1±1.9 | - | 202 | 195 | 202 | - | 202 | 195 | 202 | - |
| SLDD-Model | 84.5±0.2 | 91.1±0.1 | 75.6±0.2 | 72.7±0.0 | **50** | **50** | **50** | **50** | **5** | **5** | **5** | **5** |
| QPM (Ours) | 85.1±0.3 | 91.8±0.3 | **75.7**±0.8 | 74.2±0.0 | **50** | **50** | **50** | **50** | **5** | **5** | **5** | **5** |
| $n_{\mathrm{wc}}=10$ (Ours) | **85.7**±0.2 | **92.1**±0.1 | 75.2±0.6 | **74.5**±0.1 | **50** | **50** | **50** | **50** | 10 | 10 | 10 | 10 |

Table 14: Comparison on Interpretability metrics with Resnet50. Due to required annotations, Structural Grounding can only be computed for TravelingBirds and CUB-2011. Among more interpretable models, the best result is marked in bold, second best underlined.

| Method | SID@5↑ | | | | Class-Independence↑ | | | | Contrastiveness↑ | | | | Structural Grounding↑ | |
|---|---|---|---|---|---|---|---|---|---|---|---|---|---|---|
| | CUB | CARS | TRAVEL | IMGNET | CUB | CARS | TRAVEL | IMGNET | CUB | CARS | TRAVEL | IMGNET | CUB | TRAVEL |
| Baseline Resnet50 | 57.7±0.4 | 54.4±0.3 | 59.9±0.4 | 37.1 | 98.0±0.0 | 97.8±0.0 | 98.0±0.0 | 99.4 | 74.4±0.1 | 75.1±0.1 | 74.4±0.1 | 71.6 | 34.0±0.3 | 32.1±0.2 |
| glm-saga$_5$ | 55.4±0.5 | 51.8±0.3 | 56.0±0.8 | 35.8±0.0 | 97.8±0.0 | 97.6±0.0 | 97.8±0.0 | 99.4±0.0 | 74.0±0.1 | 74.5±0.1 | 73.8±0.1 | 71.7±0.0 | 2.5±1.0 | 4.4±2.8 |
| PIP-Net | **99.2**±0.1 | **99.0**±0.1 | **98.7**±0.1 | - | 75.6±0.4 | 62.9±0.1 | 73.4±0.1 | - | **99.6**±0.0 | **99.7**±0.0 | **99.7**±0.0 | - | 6.7±0.9 | 6.9±1.3 |
| ProtoPool | 24.5±0.8 | 30.7±3.4 | 31.5±1.6 | - | 96.9±0.1 | 96.0±0.5 | 95.5±0.1 | - | 76.7±1.0 | 78.9±2.0 | 85.2±0.5 | - | 13.9±0.9 | 7.6±2.5 |
| SLDD-Model | 88.2±0.2 | 88.6±0.6 | 87.5±0.4 | 64.7±0.7 | 96.2±0.1 | 95.5±0.1 | 96.5±0.1 | 98.6±0.0 | 87.3±0.2 | 89.7±0.3 | 86.3±0.2 | **93.4**±0.1 | 29.2±4.0 | 30.7±3.1 |
| QPM (Ours) | 90.1±0.3 | 89.6±0.4 | 89.7±0.2 | 64.1±0.7 | 97.0±0.0 | 96.5±0.0 | 97.2±0.0 | 99.1±0.0 | 96.0±0.4 | 97.7±0.4 | 94.0±0.3 | 89.3±0.1 | 47.9±2.7 | 54.3±4.0 |
| $n_{wc} = 10$ (Ours) | 95.8±0.7 | 96.6±0.5 | 95.1±0.3 | **80.1**±0.9 | **98.1**±0.0 | **98.0**±0.1 | **98.1**±0.0 | **99.5**±0.0 | 95.9±0.5 | 98.6±0.2 | 94.2±0.3 | 87.4±0.1 | **52.3**±1.6 | **62.9**±2.8 |

Table 15: Comparison on compactness and accuracy with Resnet34: QPM shows increased accuracy and compactness. Among more interpretable models, the best result is marked in bold, second best underlined.

| Method | Accuracy ↑ | | | Total Features ↓ | | | Features / Class ↓ | | |
|---|---|---|---|---|---|---|---|---|---|
| | CUB | CARS | TRAVEL | CUB | CARS | TRAVEL | CUB | CARS | TRAVEL |
| Baseline Resnet34 | 85.7±0.3 | 91.5±0.2 | 61.3±0.4 | 2048 | 2048 | 2048 | 2048 | 2048 | 2048 |
| glm-saga$_5$ | 72.0±1.0 | 82.0±0.6 | 53.5±0.8 | 442±5 | 453±6 | 443±6 | **5** | **5** | **5** |
| SLDD-Model | 83.2±0.3 | 90.7±0.3 | 74.0±0.2 | **50** | **50** | **50** | **5** | **5** | **5** |
| QPM (Ours) | 83.0±0.2 | 91.3±0.0 | 75.1±0.3 | **50** | **50** | **50** | 5 | 5 | 5 |
| $n_{\text{wc}} = 10$ (Ours) | **83.9**±0.1 | **91.7**±0.1 | **75.7**±0.6 | **50** | **50** | **50** | 10 | 10 | 10 |

Table 16: Comparison on Interpretability metrics with Resnet34. Due to required annotations, Structural Grounding can only be computed for TravelingBirds and CUB-2011. Among more interpretable models, the best result is marked in bold, second best underlined.

| Method | SID@5 ↑ | | | Class-Independence ↑ | | | Contrastiveness ↑ | | | Structural Grounding ↑ | |
|---|---|---|---|---|---|---|---|---|---|---|---|
| | CUB | CARS | TRAVEL | CUB | CARS | TRAVEL | CUB | CARS | TRAVEL | CUB | TRAVEL |
| Baseline Resnet34 | 62.1±0.3 | 56.6±0.4 | 64.1±0.6 | 97.9±0.0 | 97.7±0.0 | 98.0±0.0 | 76.4±0.1 | 77.9±0.2 | 76.0±0.1 | 39.6±0.2 | 36.0±0.4 |
| glm-saga$_5$ | 59.9±0.4 | 55.3±0.3 | 60.6±0.4 | **97.9**±0.0 | 97.7±0.0 | 97.9±0.0 | 76.5±0.0 | 77.8±0.2 | 76.0±0.1 | 7.6±2.2 | 9.4±3.8 |
| SLDD-Model | 90.1±0.8 | 86.7±2.5 | 87.3±0.3 | 97.5±0.0 | 97.6±0.2 | 97.8±0.0 | 86.0±1.0 | 83.3±4.6 | 82.0±1.4 | 24.5±2.7 | 29.4±5.2 |
| QPM (Ours) | 90.5±0.5 | 89.1±1.1 | 89.7±0.7 | 97.5±0.0 | 96.9±0.1 | 97.6±0.0 | 95.5±0.2 | 94.7±1.1 | 94.3±0.5 | 39.0±2.9 | 49.7±4.7 |
| $n_{wc} = 10$ (Ours) | **97.0**±0.3 | **98.1**±0.5 | **98.3**±0.0 | 97.9±0.1 | **98.4**±0.0 | 96.1±0.1 | **96.9**±0.3 | **98.7**±0.5 | **95.4**±0.2 | **54.7**±3.8 | **60.3**±2.0 |

Table 17: Comparison on compactness and accuracy with Inception-v3: QPM shows increased accuracy and compactness. Among more interpretable models, the best result is marked in bold, second best underlined.

| Method | Accuracy↑ | | | Total Features↓ | | | Features / Class↓ | | |
|---|---|---|---|---|---|---|---|---|---|
| | CUB | CARS | TRAVEL | CUB | CARS | TRAVEL | CUB | CARS | TRAVEL |
| Baseline Inception-v3 | 86.1±0.1 | 92.6±0.2 | 68.7±0.4 | 2048 | 2048 | 2048 | 2048 | 2048 | 2048 |
| glm-saga$_5$ | 79.2±0.5 | 89.3±0.3 | 63.4±0.5 | 814±9 | 795±8 | 813±9 | **5** | **5** | **5** |
| SLDD-Model | 83.1±0.4 | 91.1±0.2 | 69.9±0.2 | **50** | **50** | **50** | **5** | **5** | **5** |
| QPM (Ours) | 84.2±0.4 | **91.7**±0.1 | **71.5**±0.4 | **50** | **50** | **50** | **5** | **5** | **5** |
| $n_{\mathrm{wc}} = 10$ (Ours) | **84.4**±0.4 | **91.7**±0.2 | 70.8±0.3 | **50** | **50** | **50** | 10 | 10 | 10 |

Table 18: Comparison on Interpretability metrics with Inception-v3. Due to required annotations, Structural Grounding can only be computed for TravelingBirds and CUB-2011. Among more interpretable models, the best result is marked in bold, second best underlined.

| Method | SID@5 ↑ | | | Class-Independence ↑ | | | Contrastiveness ↑ | | | Structural Grounding ↑ | |
| --- | --- | --- | --- | --- | --- | --- | --- | --- | --- | --- | --- |
| | CUB | CARS | TRAVEL | CUB | CARS | TRAVEL | CUB | CARS | TRAVEL | CUB | TRAVEL |
| Baseline Inception-v3 | 38.9±0.3 | 33.1±0.2 | 40.7±0.4 | 96.1±0.0 | 95.7±0.0 | 95.9±0.0 | 89.6±0.2 | 91.7±0.2 | 89.8±0.1 | 7.1±9.6 | 24.1±0.3 |
| glm-saga$_5$ | 39.3±0.2 | 34.0±0.4 | 41.0±0.3 | 95.4±0.0 | 95.0±0.0 | 95.3±0.0 | 91.3±0.3 | 93.4±0.2 | 91.2±0.1 | 0.3±0.4 | 2.5±2.8 |
| SLDD-Model | **58.1±1.2** | **52.1±1.5** | **60.5±1.3** | 92.6±0.1 | 92.1±0.1 | 92.6±0.2 | 93.0±0.3 | **94.4±0.2** | 92.3±0.3 | 24.4±2.3 | 27.2±5.2 |
| QPM (Ours) | 48.6±0.9 | 42.8±0.8 | 50.2±0.4 | 95.1±0.1 | 94.7±0.0 | 95.1±0.1 | **93.4±0.1** | 94.3±0.1 | **93.4±0.1** | 34.8±3.4 | 44.3±3.8 |
| $n_{\text{wc}} = 10$ (Ours) | 54.6±0.6 | 47.1±0.5 | 55.0±1.4 | **96.9±0.1** | **96.8±0.0** | **96.9±0.1** | 92.6±0.2 | 93.6±0.2 | 92.6±0.2 | **41.9±3.0** | **50.3±1.6** |

Table 19: Comparison on compactness and accuracy with Swin Transformer small: QPM shows increased accuracy and compactness. Among more interpretable models, the best result is marked in bold, second best underlined.

| Method | Accuracy↑ | | | Total Features↓ | | | Features / Class↓ | | |
|---|---|---|---|---|---|---|---|---|---|
| | CUB | CARS | TRAVEL | CUB | CARS | TRAVEL | CUB | CARS | TRAVEL |
| Baseline Swin Transformer small | 87.0±0.1 | 90.6±0.6 | 59.0±0.4 | 768 | 768 | 768 | 768 | 768 | 768 |
| glm-saga$_5$ | 76.5±0.4 | 75.5±1.2 | 46.4±1.0 | 572±4 | 559±8 | 561±11 | **5** | **5** | **5** |
| SLDD-Model | 85.3±0.4 | **89.1**±0.7 | 60.3±1.0 | **50** | **50** | **50** | **5** | **5** | **5** |
| QPM (Ours) | 85.0±0.4 | 88.7±0.5 | **61.0**±1.0 | **50** | **50** | **50** | **5** | **5** | **5** |
| $n_{\mathrm{wc}} = 10$ (Ours) | **85.4**±0.3 | 89.0±0.8 | 60.9±1.1 | **50** | **50** | **50** | 10 | 10 | 10 |

Table 20: Comparison on Interpretability metrics with Swin Transformer small. Due to required annotations, Structural Grounding can only be computed for TravelingBirds and CUB-2011. Among more interpretable models, the best result is marked in bold, second best underlined.

| Method | SID@5↑ | | | Class-Independence↑ | | | Contrastiveness↑ | | | Structural Grounding↑ | |
|---|---|---|---|---|---|---|---|---|---|---|---|
| | CUB | CARS | TRAVEL | CUB | CARS | TRAVEL | CUB | CARS | TRAVEL | CUB | TRAVEL |
| Baseline Swin Transformer small | 26.4±0.1 | 26.0±0.1 | 29.6±0.2 | 96.8±0.0 | f96.6±0.0 | 96.9±0.1 | 98.3±0.1 | 98.8±0.1 | 97.4±0.1 | 24.5±0.4 | 23.6±0.9 |
| glm-saga$_5$ | 26.4±0.2 | 26.1±0.1 | 30.0±0.4 | 96.6±0.0 | 96.4±0.0 | 96.7±0.0 | **99.1**±0.1 | **99.6**±0.0 | 98.3±0.1 | 8.8±2.8 | 6.9±1.5 |
| SLDD-Model | 38.0±0.5 | 35.6±1.1 | 43.2±1.4 | 93.4±0.1 | 93.3±0.2 | 93.6±0.2 | 99.0±0.2 | 99.4±0.2 | **99.3**±0.4 | 37.2±3.4 | 40.6±2.4 |
| QPM (Ours) | 33.6±0.4 | 32.0±0.3 | 37.9±0.7 | 95.2±0.0 | 94.7±0.0 | 95.2±0.1 | 98.5±0.3 | 99.1±0.2 | 99.1±0.3 | 45.1±3.2 | 43.1±3.4 |
| $n_{wc} = 10$ (Ours) | **40.4**±0.7 | **37.3**±1.7 | **44.6**±1.2 | **96.9**±0.0 | **96.6**±0.0 | **96.9**±0.1 | 95.5±0.3 | 97.7±0.6 | 94.7±1.0 | **52.1**±2.1 | **52.5**±2.0 |

Table 21: Comparison on compactness and accuracy with Swin Transformer tiny: QPM shows increased accuracy and compactness. Among more interpretable models, the best result is marked in bold, second best underlined.

| Method | Accuracy ↑ | | | Total Features↓ | | | Features / Class↓ | | |
|---|---|---|---|---|---|---|---|---|---|
| | CUB | CARS | TRAVEL | CUB | CARS | TRAVEL | CUB | CARS | TRAVEL |
| Baseline Swin Transformer tiny | 86.6±0.2 | 90.3±0.3 | 60.1±0.6 | 768 | 768 | 768 | 768 | 768 | 768 |
| glm-saga$_5$ | 71.8±2.0 | 70.8±1.4 | 46.6±1.5 | 559±15 | 559±9 | 569±10 | **5** | **5** | **5** |
| SLDD-Model | 84.4±0.3 | **88.4**±1.3 | 58.3±0.7 | **50** | **50** | **50** | **5** | **5** | **5** |
| QPM (Ours) | 84.5±0.4 | 88.0±1.3 | 59.3±0.9 | **50** | **50** | **50** | 10 | 10 | 10 |
| $n_{\mathrm{wc}} = 10$ (Ours) | **84.6**±0.4 | 88.3±1.0 | **59.8**±0.9 | **50** | **50** | **50** | 10 | 10 | 10 |

Table 22: Comparison on Interpretability metrics with Swin Transformer tiny. Due to required annotations, Structural Grounding can only be computed for TravelingBirds and CUB-2011. Among more interpretable models, the best result is marked in bold, second best underlined.

| Method | SID@5↑ | | | Class-Independence ↑ | | | Contrastiveness↑ | | | Structural Grounding↑ | |
|---|---|---|---|---|---|---|---|---|---|---|---|
| | CUB | CARS | TRAVEL | CUB | CARS | TRAVEL | CUB | CARS | TRAVEL | CUB | TRAVEL |
| Baseline Swin Transformer tiny | 27.3±0.2 | 26.3±0.1 | 30.9±0.2 | 96.8±0.0 | 96.6±0.0 | 96.9±0.0 | 98.8±0.0 | 99.0±0.1 | 98.4±0.1 | 26.1±0.2 | 23.1±0.4 |
| glm-saga$_5$ | 26.4±0.2 | 25.9±0.1 | 29.9±0.3 | 96.6±0.0 | **96.5**±0.0 | 96.7±0.0 | **99.3**±0.0 | **99.5**±0.1 | **98.9**±0.1 | 12.1±3.9 | 6.1±2.2 |
| SLDD-Model | 38.9±0.6 | 35.4±1.6 | 46.6±0.7 | 93.3±0.2 | 92.9±0.1 | 93.6±0.2 | 99.2±0.3 | 99.3±0.4 | **98.9**±0.2 | 43.8±5.7 | 41.2±1.4 |
| QPM (Ours) | 36.9±0.4 | 31.5±1.4 | 41.8±0.3 | 95.1±0.1 | 94.6±0.1 | 95.3±0.1 | 98.2±0.4 | 98.7±0.4 | 98.4±0.2 | 50.9±4.9 | 51.4±2.1 |
| $n_{wc} = 10$ (Ours) | **44.6**±0.5 | **37.5**±0.4 | **48.5**±0.9 | **96.9**±0.0 | **96.5**±0.0 | **97.0**±0.0 | 93.8±0.2 | 97.4±0.8 | 92.8±0.7 | **54.7**±3.8 | **54.5**±2.6 |

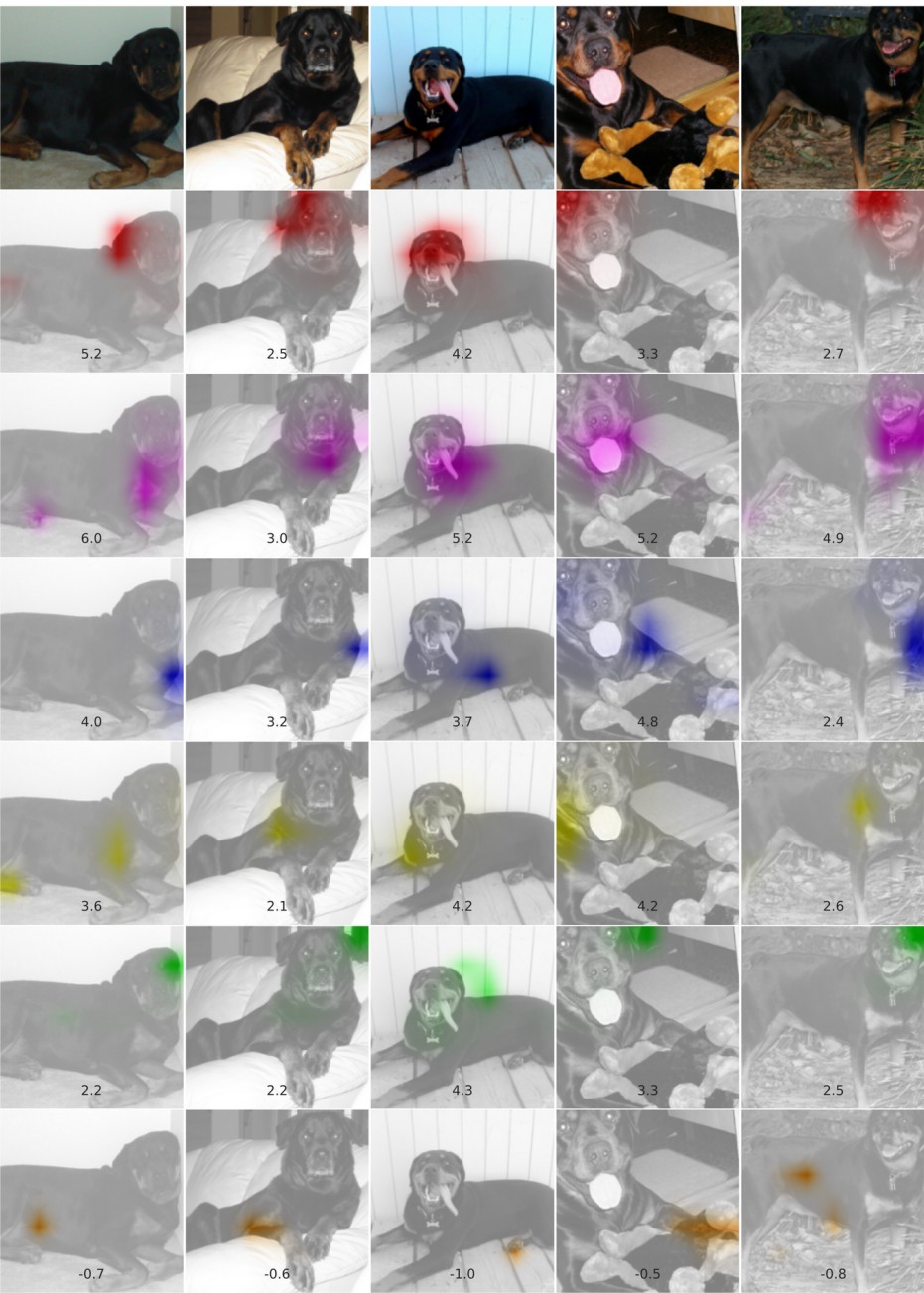

Figure 21: Exemplary Activations of Features in fig. 1 on further Rottweiler images. The feature values after normalization are written on the images. Note that all shown activations are scaled from 0 to 1, resulting in an arbitrary localization of the brown feature detecting the Doberman-like head.

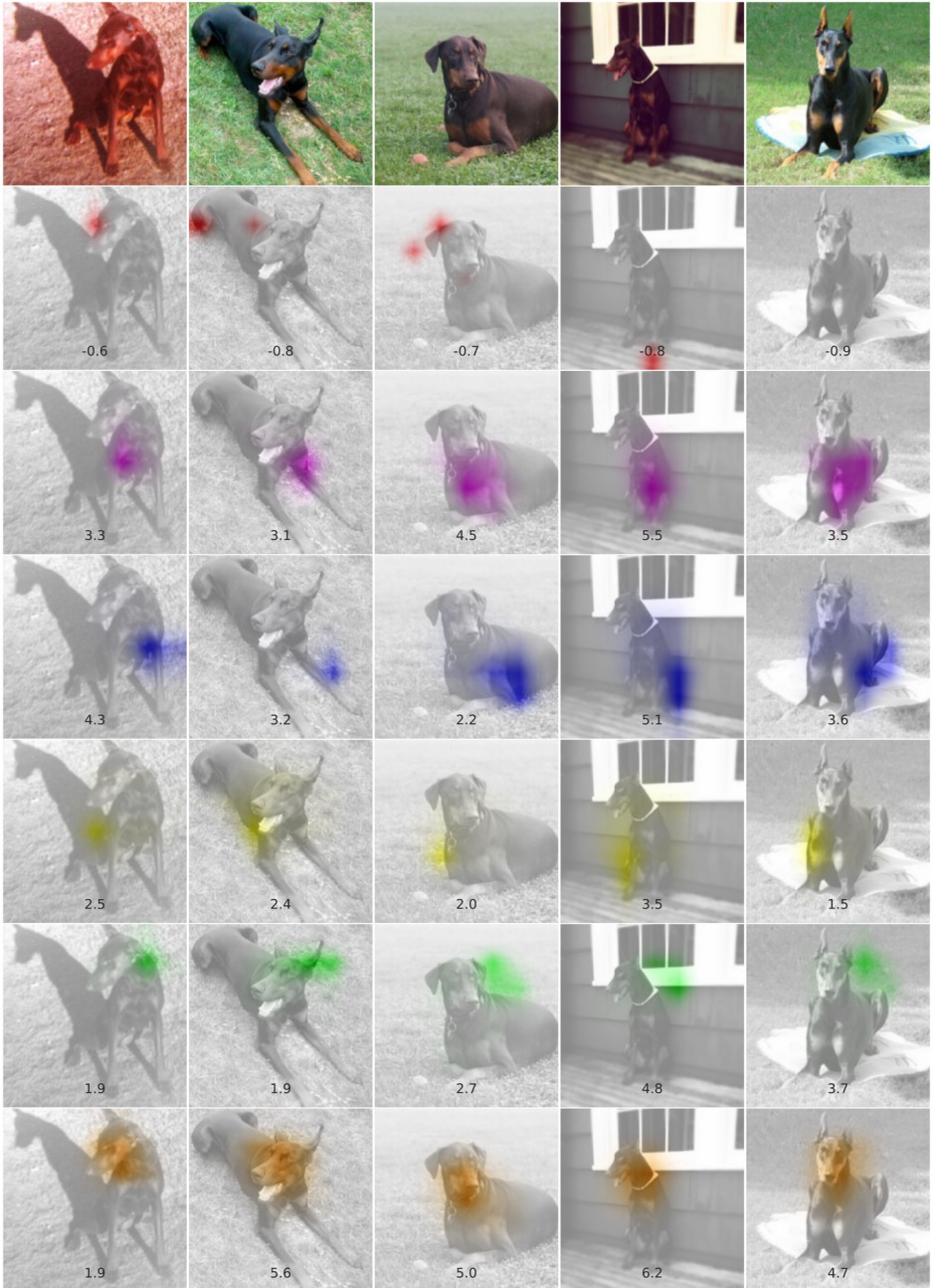

Figure 22: Exemplary Activations of Features in fig. 1 on further Doberman images. The rounded feature values after normalization are written on the images. Note that all shown activations are scaled from 0 to 1, resulting in an arbitrary localization of the red feature detecting the Rottweiler-like head.

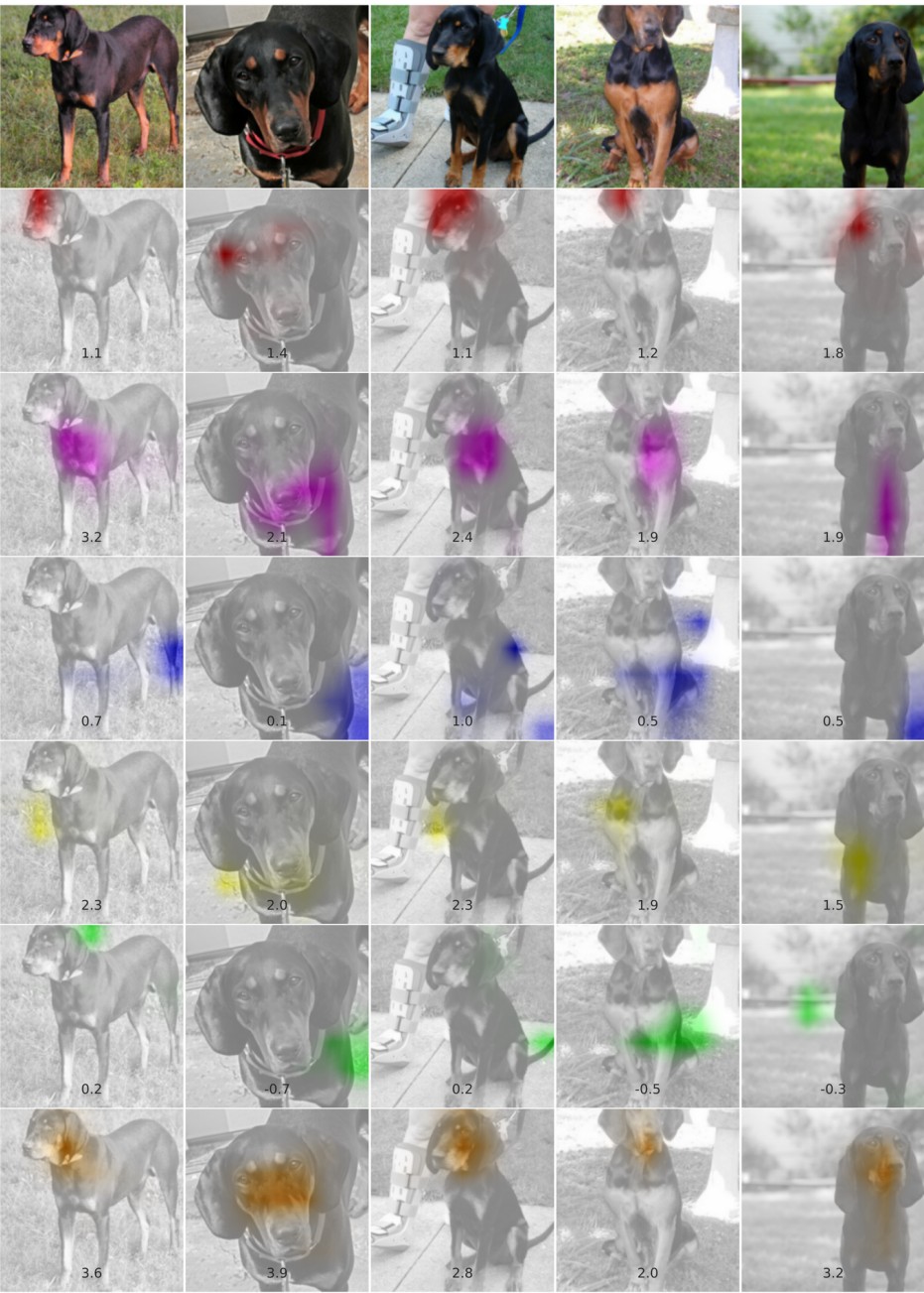

Figure 23: Exemplary Activations of Features in fig. 1 on Black and Tan Coonhound images. The rounded feature values after normalization are written on the images. Note that all shown activations are scaled from 0 to 1, resulting in an arbitrary localization of the two not assigned and barely activated blue and green features. The fifth assigned feature is shared with dog types such as Newfoundlands, bears and pandas, localizing on the neck region.

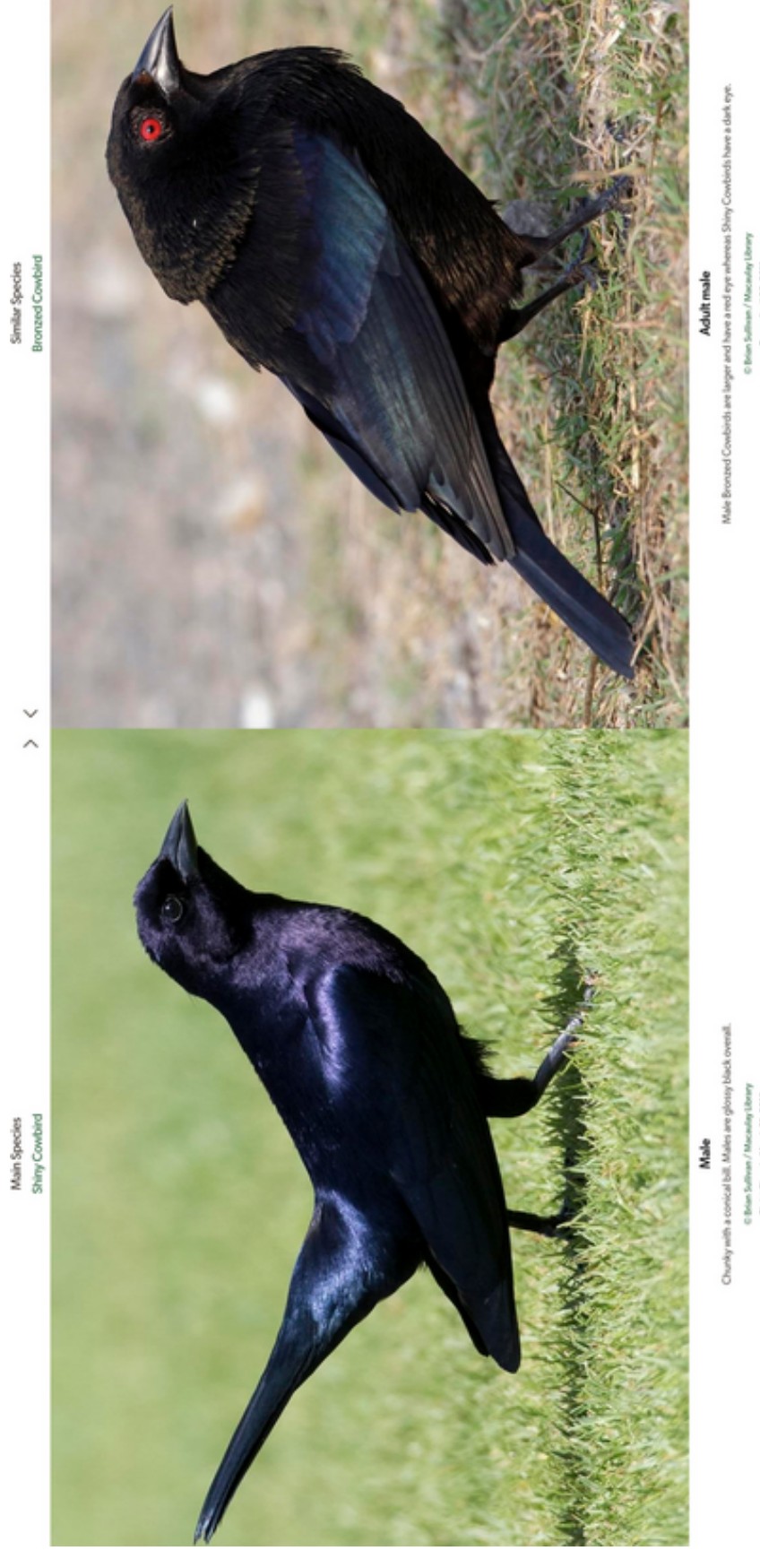

Figure 24: Screenshot of the Cornell-Lab website (Sullivan, 2024), describing how the similar species shiny and bronzed cowbird differ: The only differences explained are in the size, which is not usable for our QPM due to cropping, and the eye color, which our QPM learned without any supervision and explained in fig. 4 as the differentiating factor.

