# OpenReview forum: "QPM: Discrete Optimization for Globally Interpretable Image Classification"
_ICLR.cc/2025/Conference — ICLR 2025 Poster_

### Official Review · Reviewer_GMvn · 2024-11-03

**Soundness:** 4
**Presentation:** 3
**Contribution:** 3
**Rating:** 8
**Confidence:** 4

**Summary:**

With the rise of AI in several high-risk areas such as autonomous driving and medical diagnosis, being able to interpret the decisions of deep neural networks is paramount for safety. The authors focus on providing faithful global explanations as opposed to the simpler task of giving local explanations. This is achieved via their Quadratic Programming Enhanced Model (QPM) via learning interpretable class representations using only 5 features and binary assignment. This simple decision logic makes interpreting classifications feasible by humans. The authors optimize for this representation and predictive power using discrete optimization on diverse pretrained features. They evaluate their method on a variety of datasets and achieve competitive accuracy and interpretability compared to popular interpretability methods. The authors show qualitative results of their model’s interpretability and quantitative for metrics on complexity, spatial diversity, class-indepence of features, contrastiveness of features, and their structural grounding to human given concepts.

**Strengths:**

- Global explanations are simple and easy for humans to interpret. Weights from features are binary and only positive. Each class only has 5 features (in most experiments).
- Robustness and generalizability of their method is given based on evaluation on a variety of datasets and models
- Good use of discrete optimization and constraints to select diverse, localized, and predictive features from a pre trained set of features. Clever relaxation of constraints for feasible optimization while maintaining accuracy and interpretability.
- Ability for concepts/features to be used for predicting multiple classes.
- Competitive or SOTA accuracy and interpretability compared to other popular methods. Qualitative results show the ease of interpretability of their method.
- Ablation experiments show effectiveness of $Z_B$ and $Z_R$ objectives.

**Weaknesses:**

Minor:
- Writing can be unclear at times:
  - L282: Are you keeping the entries that are being clipped? Between 0 and epsilon.
  - L127-129: This sentence is unclear. How does the balanced binary assignment cause general binary concepts?
  - L372-273: You say you omit measuring the grounding of features, but provide results…Do you mean that you don’t focus on it since you only do this for CUB & TravelingBird due to lack of part annotations in other datasets?

Worth noting:
- While your method doesn’t use negative reasoning, which makes it more simple, it is a restriction/limitation.
- Your method’s performance isn’t that much above Q-SENN, but you mention the computation limitation on Q-SENN. A comparative analysis, such as a table, that shows the speed of training differences between methods would be desirable.

**Questions:**

- You provide misclassifications of your model in the Appendix in figure 10, but not the explanation for the classification visually. I think being able to understand the misclassification from your method’s explanation is important to your method’s strength. Could you show the explanation visualization for several misclassified samples? I think image (j) would be interesting plus images from other datasets.
- See weaknesses for other questions.

---

> ### Author Response · Authors · 2024-11-18
> **Question regarding Line 282**
>
> We thank you for your valuable feedback and acknowledging several strengths. We thank you especially for appreciating providing global explanations as more challenging than local explanations, and considering ours as easy for humans to interpret. We respond to all weaknesses / questions separately.
>
> > L282: Are you keeping the entries that are being clipped? Between 0 and epsilon.
>
> Thanks for pointing to a confusing sentence. What do you specifically mean with “keep”? \
> The values between 0 and epsilon are set to 0. When passing this matrix to Gurobi, the zeros are effectively discarded for the optimization, which we will clarify in section 4.1.1.

---

> ### Author Response · Authors · 2024-11-18
> **Clarification Regarding Impact of Balanced Binary Assignment**
>
> > L127-129: This sentence is unclear. How does the balanced binary assignment cause general binary concepts?
>
> Thanks for pointing out this unclear formulation. We meant that due to every class being represented by the same number of features, each of them is not forced to detect the class on its own, leading to more general features and improved generalization and accuracy (and indeed also more Contrastiveness of the features). \
> We added an ablation to new appendix G and table 7, that evaluates a model without this balance constraint. This then harms the overall accuracy of the model (-0.8 on CUB, -0.2 Cars). We further support our argument by showing the declining Class-Independence of the assigned features based on the sparsity of the assigned classes in new Figure 16.
>
> Finally, we will phrase this sentence more clearly: \
> As every class is assigned to the same number of features, each of the features detects shared general concepts between its assigned classes instead of also detecting the entire class. This leads to state-of-the-art accuracy.

---

> ### Author Response · Authors · 2024-11-18
> **Clarification Regarding Measuring the Grounding of Features**
>
> > L372-273: You say you omit measuring the grounding of features, but provide results…Do you mean that you don’t focus on it since you only do this for CUB & TravelingBird due to lack of part annotations in other datasets?
>
> We do not measure the grounding of individual features, as polysemantic features make that very difficult to quantify.  \
> Instead, we measure Structural Grounding, which measures if similar classes in reality are also similar in our model. This bypasses the problem of polysemantic features, as similarity is measured across features, which will typically only activate on one concept depending on their jointly predicted features. E.g. the “red eye” feature from Fig. 4 will most likely only respond to the red eye in all pictures of cowbirds, so that a misdetection does not influence the loss. \
> To support and discuss this hypothesis of local monosemantic features, we added new Appendix H alongside figure 18 and will add a discussion on it to the appendix. Figure 18 demonstrates that around a given class, a feature can be effectively aligned with just one attribute, in this case the red eye. \
> How to robustly quantify the alignment of polysemantic features will also be added as an avenue for future work.

---

> ### Author Response · Authors · 2024-11-18
> **Clarification Regarding Negative Reasoning**
>
> > While your method doesn’t use negative reasoning, which makes it more simple, it is a restriction/limitation.
>
> You are right, that this is a restriction of the QP in its current form. \
> While we believe that it is generally preferable to represent classes only using positive assignments, we agree that one can think of other datasets where negative reasoning is more helpful. We add this as an avenue for future work.

---

> ### Author Response · Authors · 2024-11-18
> **Clarification Regarding Runtime Compared to Q-SENN**
>
> > Your method’s performance isn’t that much above Q-SENN, but you mention the computation limitation on Q-SENN. A comparative analysis, such as a table, that shows the speed of training differences between methods would be desirable.
>
> Thank you for your suggestion. We added new table 6 and appendix F, which shows that Q-SENN takes more time on ImageNet due to its iterative optimization with more training epochs, whereas solving the QP takes relatively longer on the small (in terms of samples) dataset CUB.  This is due to the QP runtime scaling with number of features to select and initial and number of classes. In contrast Q-SENN with its SGD-like fitting primarily scales with the number of training samples to iterate over, causing longer runtime on the larger ImageNet.
>
> We want to emphasize that one of the main improvements over Q-SENN lies in the contrastive faithfully interpretable class representations, enabling far superior interpretability.

---

> ### Author Response · Authors · 2024-11-18
> **Question Regarding Explanation Visualization for Misclassified Samples**
>
> > You provide misclassifications of your model in the Appendix in figure 10, but not the explanation for the classification visually. I think being able to understand the misclassification from your method’s explanation is important to your method’s strength. Could you show the explanation visualization for several misclassified samples? I think image (j) would be interesting plus images from other datasets.
>
> Thank you for your suggestion. We gladly add visualizations to the appendix.  \
> We included GradCAMs of the images in fig.10 in new fig.11., demonstrating that the model indeed focuses on the dogs. As QPM enables more in-depth analysis, we additionally included visualizations for the predicted class Greater Swiss Mountain Dog with its features in new fig. 12, and for the annotated class Rottweiler in new fig. 13. As you asked for another dataset, we also added new fig. 14, containing failure cases and correct classifications for the probed model in fig. 4.
> They once again highlight that the model behaves as one would expect given the global explanation: When e.g. the red eye is not present, the bronzed Cowbird is not recognized, as that feature is not found.
>
> Overall, as one significant benefit from QPM lies in being able to predict how the model will work on a test instance given its class explanation, we focused on the class explanations already presented.

---

> ### Comment · Reviewer_GMvn · 2024-11-26
>
> Thank you for the clarifications and modifications! I will still keep my rating as it stands (8: accept, good paper).

---

> > ### Author Response · Authors · 2024-11-27
> > **Official Comment by Authors**
> >
> > Dear Reviewer GMvn, we would like to thank you for your encouraging and positive feedback to our submission and response. We appreciate that you maintain the score and still see our submission as clearly being acceptable for ICLR.

---

### Official Review · Reviewer_x2rh · 2024-11-04

**Soundness:** 3
**Presentation:** 3
**Contribution:** 3
**Rating:** 6
**Confidence:** 4

**Summary:**

To develop inherently interpretable image classification, the authors propose to use a set of five diverse features for each class to perform classification. Specifically, the proposed method leverages discrete quadratic programming to extract the features that are distinct, contrastive, and generalizable across similar classes from pretrained vision models, e.g., ResNet-50. The paper provides extensive empirical results on prediction accuracy and interpretability metrics.

**Strengths:**

1. The idea is intuitive and easy to understand.
2. The paper is well-organized and easy to follow.
3. Different from most interpretable models, the proposed method didn’t compromise too much on the prediction accuracy.

**Weaknesses:**

1. Although the proposed method can provide five different heatmaps to explain the prediction of a class, the semantic meanings of these explanations are subjective. This is different from Concept Bottleneck Models (CBMs) and Testing with Concept Activation Vectors (TCAV) that can provide human-understandable concept explanations with semantic meanings. Although the authors propose structural grounding that uses dense annotations of bird parts from the CUB dataset to interpret some of the heatmaps, the concept coverage might be restrictive and cannot reflect what the model has learned.
2. The choice of five features per class might be arbitrary. Some classes might be more complicated than others, while some classes could be very simple. Would it be better to use a different or dynamic number of concepts for each class? This is actually related to the Weakness 1 above. The restricted number of five concepts makes the model learn polysemantic features, which means it might represent different concepts within a single feature. This can make the interpretation of the functionality of the features even harder. For example, as shown in Fig. 12, the green feature can highlight bird beaks on one image and wings on the other image. This might defeat the purpose of making the model more interpretable.
3. The proposed method highly relies on the quality of extracted features from the pretrained models, whereas they are not designed to be explainable. Since the base models are normally trained, they can also extract biased features which would significantly affect the performance of the proposed model.
4. Experiments using standard metrics for explanations are missing, such as the explanation fidelity using the deletion/insertion method [1]. The proposed contrastiveness metric cannot fully reflect the explanation quality in terms of faithfulness. The authors propose Scale-Invariant-Diversity@5 (SID@5), however, the proposed method cannot achieve the best or second-best performance on the three datasets.

[1] Petsiuk, V. "Rise: Randomized Input Sampling for Explanation of black-box models." BMVC 2018.

**Questions:**

See Weakness for details.

---

> ### Author Response · Authors · 2024-11-18
> **Clarification regarding Semantic Meaning of Learnt features**
>
> We thank you for your valuable feedback and acknowledging several strengths, such as the extensive evaluations with relatively high predictive accuracy. We respond to all weaknesses / questions separately.
>
> > Although the proposed method can provide five different heatmaps to explain the prediction of a class, the semantic meanings of these explanations are subjective. This is different from Concept Bottleneck Models (CBMs) and Testing with Concept Activation Vectors (TCAV) that can provide human-understandable concept explanations with semantic meanings. Although the authors propose structural grounding that uses dense annotations of bird parts from the CUB dataset to interpret some of the heatmaps, the concept coverage might be restrictive and cannot reflect what the model has learned.
>
> * Thanks for pointing out that we left out the semantic alignment of the learned features. \
> This paper introduces QPM, a model that is more interpretable by design. Similar to Q-SENN, the learned features, which are more interpretable due to the structure of the final layer, can be aligned post-hoc, using e.g. the mentioned TCAV, the proposed methods from SLDD or Q-SENN, or, if no annotations are available, Clip-Dissect ( https://arxiv.org/abs/2204.10965). \
> Additionally, due to the easy-to-interpret assignment, there is more to gain from aligning individual features and the built-in notion of locality can further help with the alignment, even with polysemantic features, as shown in new Figure 18. We will add this to the related Work and discuss it in the appendix.
>
> * Concept Bottleneck models learn a concept layer based on annotated concepts, which makes it not broadly applicable. When training the classifier and concept predictor independently, their accuracy is significantly worse, indicating that just the annotated concepts might not be sufficient. \
> Further, when trained jointly, the concept predictor is actually predicting a lot more than just the concepts, making them lose significant interpretability and semantic meaning (https://arxiv.org/abs/2112.03184, https://arxiv.org/abs/2105.04289). \
> To conclude QPM differs by design from CBMs  by not being limited by the set of annotated concepts, being more accurate and learns more interpretable contrastive binary class representations, which are composed of features, that are in both cases not perfectly aligned with just one annotated concept.
>
> * We want to clarify that we do not use Structural Grounding to align single features with attributes. \
> Instead, we quantify how well the learned representations represent reality based on their similarity to other classes, a notion that Reviewer vzLV considers to be crucial.  \
> As explained in lines 370-372, we do not generally align individual features with concepts due to their polysemantic nature, as also discussed in the following point. This point is further discussed in new appendix H.

---

> ### Author Response · Authors · 2024-11-18
> **Clarification Regarding Choice of Even Distribution of Feature Assignments and Polysemanticity**
>
> > The choice of five features per class might be arbitrary. Some classes might be more complicated than others, while some classes could be very simple. Would it be better to use a different or dynamic number of concepts for each class? This is actually related to the Weakness 1 above. The restricted number of five concepts makes the model learn polysemantic features, which means it might represent different concepts within a single feature. This can make the interpretation of the functionality of the features even harder. For example, as shown in Fig. 12, the green feature can highlight bird beaks on one image and wings on the other image. This might defeat the purpose of making the model more interpretable.
>
> * The number of features per class is indeed a hyperparameter, but five is not quite chosen arbitrarily, as it can be considered low enough for every human to follow (Miller, 1956) and a lower value leads to more contrastable representations. \
> Additionally, the ablation in Fig. 7 shows that it is sufficient on those datasets. However, it can indeed be optimized for other datasets.  Table 8 (revised 13) further shows that the gains on ImageNet with 10 per class are also marginal.
>
> * You are right that using a dynamic number of features per class can make sense, if they consist of a varying number of concepts. However, there are some things to consider:
> An even assignment of features leads to exclusively comparable class representations, with no bias or other scaling harming global interpretability, as shown in e.g. Fig. 1 or 4. Additionally, the lower the number of features assigned to one class, the more likely the assigned features to detect the class alongside a shared concept, harming their generality. This then harms the overall accuracy of the model (-0.8 on CUB, -0.2 Cars). We included this ablation in the new appendix G and Table. 7. \
> We further support our argument by showing the declining Class-Independence of the assigned features based on the sparsity of the assigned classes in new Figure 16.
>
> *  We agree, that polysemantic features emerge and discuss this phenomenon in new Appendix H. \
> However, we believe it is less due to the sparse assignment but rather due to the low number of total features. Further, the lack of sparsity is considered important for encoding, and especially decoding, features in superposition (https://transformer-circuits.pub/2022/toy_model/index.html). Could you elaborate on why you think high sparsity causes polysemantic features? \
> Additionally, while individual features may be polysemantic, the class representations are faithful and cause predictable outcomes. E.g. Rottweiler and Doberman are separated based on their head, and the shiny cowbird is recognized based on the eye. Due to the nature of its optimization, QPM should typically learn features that are locally monosemantic given the remaining 4 strongly activating features, as e.g. during training, the red eye feature used to distinguish the two cowbirds should only activate if the red eye is present, as the model will try to only represent other concepts with this feature, that do not occur on those images of cowbirds, so that it can satisfy its loss.  \
> This conditional alignment is unfortunately difficult to quantify holistically with the present data. Therefore, we added more interpretable failure cases in Fig. 12-14, where e.g. the Bronzed Cowbird is wrongly classified if the red eye is not present. \
> We further included anecdotal evidence for the phenomenon of local monosemanticity for the concrete feature that focuses on the red eye in Fig.4.  New Fig. 18 demonstrates the alignment, as measured by the Alignment metric from Q-SENN, of the feature that focuses on the red eye, as a function of the similarity of the label to Bronzed Cowbird. If the class is more like Bronzed Cowbird, the feature is becoming very aligned with this attribute, surpassing even CBMs alignment values (see Q-SENN paper for details).  Thus, at least for this feature, QPM seems to have learned a polysemantic, but locally monosemantic feature. \
> Nevertheless, like the green feature in Fig.12, some features can be hard to name, as models might well learn to also use non-human features, such as high-frequency patterns related to adversarial examples (https://arxiv.org/abs/1905.02175). \
> We want to emphasize, that all Neural networks usually learn polysemantic features, some of which rely on concepts not typically used by humans. However, QPM learns faithfully interpretable class representations, that lead to predictable outcomes as seen in the examples of correct classifications and failure cases, and its simple structure can even enable more advanced alignment techniques for understanding polysemantic features. \
> Polysemantic features will be added to the limitations section and robustly measuring their alignment will be added as an avenue for future work.

---

> ### Author Response · Authors · 2024-11-18
> **Clarification Regarding the Dependence on the Extracted Features prior to QP**
>
> > The proposed method highly relies on the quality of extracted features from the pretrained models, whereas they are not designed to be explainable. Since the base models are normally trained, they can also extract biased features which would significantly affect the performance of the proposed model.
>
> It is true, that the features of the dense model do not capture a concept individually.  However, they evidently already correlate with classes to which an assignment is beneficial. We further want to emphasize that we train the dense model using the Feature Diversity Loss following the SLDD-Model, which proposed and evaluated it, and Q-SENN. Thus, its features are very diversely localizing. \
> However, even without it (or as done on ImageNet), the resulting QPM is still fairly accurate and loses less than 1% accuracy.\
> Finally, we evaluated the method across 5 architectures, typically each with 5 seeds also for the dense model and see consistent good results.
>
> Regarding the biased features: As you mentioned, most of the features of the dense model do not already capture a proper concept. They more so correlate with some concepts which then emerge during fine-tuning based on their assigned classes. If that concept is biased, it will learn it but be able to communicate it faithfully. The QPM is not a model that prevents all forms of biases, even though it is robust to the biases of Travelingbirds, but rather a more interpretable model which tries to classify the dataset as effectively as possible using diverse, binary and contrastive concepts. If an on the training set effective concept is biased, the QPM explains itself and the user can decide how to move forward, which is significantly more control than the hidden biases in black-box models enable.

---

> ### Author Response · Authors · 2024-11-18
> **Clarification Regarding Choice of Metrics**
>
> > Experiments using standard metrics for explanations are missing, such as the explanation fidelity using the deletion/insertion method [1]. The proposed contrastiveness metric cannot fully reflect the explanation quality in terms of faithfulness. The authors propose Scale-Invariant-Diversity@5 (SID@5), however, the proposed method cannot achieve the best or second-best performance on the three datasets.
>
> * As appreciated by Reviewers vzLV and GMvn, we focus on providing faithful global class explanations rather than local explanations. Therefore, we do not use metrics to measure the explanation fidelity, as we do not introduce any saliency method but instead learn class representations which are by design faithful.   Holistically, one would thus have to extend the deletion/insertion methods towards evaluating the feature faithfulness. Additionally, those metrics are, while broadly and effectively used, not flawless, since removing pixels is clearly out of distribution (https://arxiv.org/abs/1806.10758). Further, our method is backbone independent and can thus even incorporate models with built-in more faithful saliency maps like B-cos (Böhle et al., 2023).
> We added this discussion as an avenue for future work to new appendix K.
>
> * Reviewer vzLV also mentioned (responded in point “Clarification on results on Interpretability Metrics”) the missing superiority in SID@5. We want to emphasize that PIP-NET cannot really be judged on this metric, as it localizes class-detections rather than concepts which we tried to communicate in text  (l. 484-486). We internally discussed adding an asterisk on the metrics of PIP-Net and $glmsaga_5$ in table 4 and then to remove their boldness, but had the feeling, that this might also distract the reader.
>
> * We additionally want to point out that the interpretability metrics do not need to be 100% for a highly interpretable model, but rather should all be high.
>   * For SID@5 specifically, it would e.g. not be per se a worse model just because it uses features that focus on the head form or color and others that focus on the eye jointly. But given the limited resolution of the feature maps, these features would have significant overlap, causing the metric to drop.
>   * As we see the challenge of self-explaining neural networks as multicriterial optimization task, we summarized the overall performance in the radarplot in figure 6 and will bring more attention to this aspect in the revised version.
> * Additionally, we want to emphasize that SID@5  refers to the diversity of the top 5 features (limiting the evaluation to classes with at least 5), a design decision made to compare across varying sparsities. But consequently, when comparing QPM and Q-SENN, it measures the spatial diversity of all features used for QPM, but only 5 from Q-SENN. Thus, when considering the sparsity distribution in new Figure 17, it is evident that the top 5 features of Q-SENN can be more diverse, as they frequently have to focus on just half of the class features.
>
> * We finally added the results on Diversity@5 in new table 8, according to which QPM has the most diverse features. We will point to it and its result of the extremely localized PIP-Net to explain the need for a scaled version.

---

> > ### Comment · Reviewer_x2rh · 2024-11-21
> >
> > I appreciate the authors' thorough responses and the detailed discussion provided in the Appendix. I find the explanations satisfactory and raise my score. If possible, I encourage the authors to consider including a portion of the extensive discussion in the main text.

---

> > > ### Author Response · Authors · 2024-11-26
> > > **Official Comment by Authors**
> > >
> > > Dear Reviewer x2rh, we would like to thank you for your encouraging and positive feedback to our submission and response. We appreciate that you raised the score and see our submission as being acceptable for ICLR. In the final revised version, we mainly add references to the added appendices, but will include some portions of them, especially the aspect of polysemantic features, in the camera-ready version.

---

### Official Review · Reviewer_vzLV · 2024-11-04

**Soundness:** 3
**Presentation:** 4
**Contribution:** 3
**Rating:** 6
**Confidence:** 3

**Summary:**

The paper introduces the Quadratic Programming Enhanced Model (QPM), an approach for globally interpretable image classification. Unlike traditional models that primarily focus on local explanations, QPM achieves a compact, interpretable global representation of each class by assigning only a few features, typically five, per class through discrete optimization. This binary feature assignment ensures contrastive and general class representations, which allows for clear, globally understandable explanations. The model sets a new benchmark in both interpretability and accuracy for compact, interpretable models across small and large datasets, including ImageNet-1K. The approach shows robust performance by using optimally selected features that enhance global interpretability and structural grounding while maintaining high classification accuracy.

**Strengths:**

- The paper studies global  an important question in explainable machine learning, providing global interpretation through user defined features.

- The paper motivation is well justified, well-written and easy to understand.

- The results show that QPM achieves superior performance in structure grounding, which signifies that QPM’s learned class representations align with human-understandable features. This is very crucial in explainable machine learning.

**Weaknesses:**

- The paper operates on only a few features, which is fine for natural images. However, I think in more complex domain applications such as medical imaging, more features might be required. How would more features affect the model’s optimization of feature selection in terms of time and accuracy?

- On that note, the paper lacks experiments related to high-stake domain applications such as healthcare, autonomous driving, etc.

- In QPM, Pearson correlation serves as a fundamental similarity measure that aids in identifying which features are most relevant to each class. By focusing on linear relationships, it allows the model to assign interpretable, contrastive, and compact features to classes. However, because it overlooks non-linear relationships, it may not capture all nuances in complex datasets, potentially impacting accuracy in applications where non-linear patterns are significant. Could the author elaborate on this matter?

- QPM demonstrates inferior performance on SID@5, Class-Independence, and Contrastiveness compared to state-of-the-art models, suggesting that its compact binary feature assignment strategy, while beneficial for interpretability, may limit the model's ability to capture diverse, independent, and highly contrastive feature representations across classes. This trade-off reflects QPM's focus on simplicity and interpretability at the expense of some nuanced, dataset-specific feature diversity.

**Questions:**

- Presentation-wise, I think it is better to explicitly show what the features are in human language for the figures, so that it is more friendly to the readers.

- Structure ground is only evaluated on CUB-2011, and the authors also explained why. How much effort would it take to achieve the evaluation metric on a larger dataset, such as ImageNet?

- How long is the optimization of “4 hours for fine-grained datasets, and roughly 11 hours for ImageNet”? Are these times better than previous methods?

---

> ### Author Response · Authors · 2024-11-18
> **Clarification on Number of Features and Medical Images**
>
> We thank you for your valuable feedback and acknowledging several strengths. We especially thank you for appreciating the value of having class representations that align with human-understandable similarities.\
> We respond to all weaknesses / questions separately.
>
> > The paper operates on only a few features, which is fine for natural images. However, I think in more complex domain applications such as medical imaging, more features might be required. How would more features affect the model’s optimization of feature selection in terms of time and accuracy?
>
> Thanks for the interesting question. Generally, as shown in Fig. 7, more features can capture more concepts and thus cause higher accuracy, with quickly diminishing returns. However, more features at a fixed number of classes also cause less interpretability as each of the features has less assigned classes and thus less guidance to become a general concept during fine-tuning. Also, with more features, less classes share a high number of features for easy comparison.  So, generally, there is a trade-off between interpretability and number of features to learn.
>
> Please also note, the optimal number of features does not define the total number of relevant concepts that should be extracted to classify the dataset, but it rather relates to how many features are needed to capture the concepts considering their interdependency and the sparse decoding of the features. This leads to   polysemanticity, where one feature responds to multiple human concepts, whose misdetection does not change the loss too much.  Notably, Q-SENN showed a dependency of the optimal number of features on the dataset. As shown in Fig. 7, QPM reaches this plateau at a lower number. We added new Appendix H and discuss this topic in more depth compared to the previous discussion around lines 370 to 372 or 169.
>
> Further, we are not aware that the more constrained medical images always show a higher number of different concepts than e.g. the natural Images in ImageNet, for which QPM performs well with just 50 features. E.g. in skin cancer, the simple rule for humans is based on just 5 abnormal concepts (ABCDE-rule: https://www.aad.org/public/diseases/skin-cancer/find/at-risk/abcdes) However, applying QPM to medical images is certainly an avenue for future work, which we will include in the appendix.\
> Notably, PIP-Net has been applied to binary medical image classification (https://arxiv.org/pdf/2307.10404) and used way less, not more than ¼ of the features needed for CUB, which is however a 200-class classification task, thus not perfectly comparable.
>
> Regarding the expected optimization time when increasing the features, the time to solve the QP scales with numbers of features to select. We added a section discussing that to appendix F, including new Fig. 15, containing the time it took to optimize for Fig.7. \
> Thus, when selecting significantly more than 50, the target gap to optimality could be slightly increased to account for that. This will decrease the time the QP is optimized for. In prior experiments we also found more suitable priors to speed up the optimization for higher number of features, but as the accuracy uplift with more features is diminishing in sync with interpretability, we did not include those, but instead compared QPM with the defaults from previous work.
>
> > On that note, the paper lacks experiments related to high-stake domain applications such as healthcare, autonomous driving, etc.
>
> As QPM learns interpretable class representations, the typical medical datasets, e.g. used in the paper above, with binary classification problems are not applicable. Similarly, we are not aware of suitable datasets from autonomous driving. However, we believe that performing well on ImageNet suggests broad applicability. We would be happy to learn about datasets from these domains that describe classification problems with many classes because we see this as ongoing research. In this submission we solely focus on basic and fundamental research, but we are happy to discuss the impact of our work on several high-stake applications in the Future Work Section.

---

> ### Author Response · Authors · 2024-11-18
> **Clarification on Pearson Correlation**
>
> > In QPM, Pearson correlation serves as a fundamental similarity measure that aids in identifying which features are most relevant to each class. By focusing on linear relationships, it allows the model to assign interpretable, contrastive, and compact features to classes. However, because it overlooks non-linear relationships, it may not capture all nuances in complex datasets, potentially impacting accuracy in applications where non-linear patterns are significant. Could the author elaborate on this matter?
>
> Thank you for your interesting question. Note that the features are not actually individually interpretable or contrastive when solving the QP. At this stage, they still encode concepts in superposition. The QP finds a good selection and assignment so that they become concepts during fine-tuning. For that, it is just important, that assigned classes are more like their feature than not assigned classes.\
> As the model is based on a linear layer between features and classes, the linear relationship of the correlation seems intuitive. However, non-linear criteria like Auroc only cause a small drop in accuracy (-0.3 or -0.2 on CUB or CARS).  We added this ablation as new Appendix J and Table 10.\
> Finally, please note that QPM is designed so that the classes are interpretably represented using the features, which capture the non-linear relationships.

---

> ### Author Response · Authors · 2024-11-18
> **Clarification on  results on Interpretability Metrics**
>
> > QPM demonstrates inferior performance on SID@5, Class-Independence, and Contrastiveness compared to state-of-the-art models, suggesting that its compact binary feature assignment strategy, while beneficial for interpretability, may limit the model's ability to capture diverse, independent, and highly contrastive feature representations across classes. This trade-off reflects QPM's focus on simplicity and interpretability at the expense of some nuanced, dataset-specific feature diversity.
>
> We thank you for pointing out that QPM’s improvement could be more pronounced on the interpretability metrics for individual features. It indeed is worse than Q-SENN and PIP-Net on SID@5, worse than $glmsaga_5$ on Class-Independence and worse than PIP-Net on Contrastiveness.
> *  We were unsure how to handle the values of PIP-Net, which cannot be properly judged, as the individual features are frequently class-detectors, which makes interpreting them as abstract human-concept obsolete.  We tried to describe this phenomenon in text (l. 484-486), but have unfortunately failed to make it clear enough.\
> Similarly, $glmsaga_5$, which can hardly be considered a state-of-the-art model considering its accuracy and missing interpretability, uses features of a dense uninterpretable model (as pointed out by x2rh).\
> We internally discussed adding an asterisk on the metrics of PIP-Net and $glmsaga_5$ in table 4 and then to remove their boldness, but had the feeling, that this might also distract the reader.
> Among models with individually more interpretable features, QPM would then be at the position 2, 1,1.\
> As we see the challenge of self-explaining neural networks as multicriterial optimization task, we summarized the overall performance in the radarplot in figure 6 and will bring more attention to this aspect in the revised version.
> * We additionally want to point out that these metrics do not need to be 100% for a highly interpretable model, but rather should all be high, which we tried to emphasize in text (lines 474-479) and Fig. 6. Notably, only QPM reaches good results on all of them.
>
>   * For SID@5 specifically, it would e.g. not be per se a worse model just because it uses features that focus on the head form or color and others that focus on the eye jointly. But given the limited resolution of the feature maps, these features would have significant overlap, causing the metric to drop.
>   * For Class-Independence, an in some way optimal Feature Extractor, that assigns a feature -value of 1 to all images that contain one of the n assigned classes and 0 else, would only get 1 – 1/n. For CUB, that would mean 95%. Everything around that value (all probed models except PIP-Net) would indicate class-independent features, whereas going higher could also be attributed to activating on features of not assigned classes.
> *  Additionally, we want to emphasize that SID@5 refers to the diversity of the top 5 features (limiting the evaluation to classes with at least 5), a design decision made to compare across varying sparsities. But consequently, when comparing QPM and Q-SENN, it measures the spatial diversity of all features used for QPM, but only 5 from Q-SENN. Thus, when considering the sparsity distribution in new figure 17, it is evident that the top 5 features of Q-SENN can be more diverse, as they frequently have to focus on just half of the class features.
> * We finally added the results on Diversity@5 in new table 8, according to which QPM has the most diverse features. We will point to it and its result of the extremely localized PIP-Net to explain the need for a scaled version.

---

> ### Author Response · Authors · 2024-11-18
> **Uncertainty about Question Regarding Describing Features in Human-Language**
>
> > Presentation-wise, I think it is better to explicitly show what the features are in human language for the figures, so that it is more friendly to the readers.
>
> We are not sure if we understand your remark. \
> Do you suggest adding text descriptions of where features localize to figures 1 and 4? \
> We did not try to name the features, as naming “what” the features are is very difficult and might hide complexity. \
> For example, the various features of the Shiny Cowbird in figure 4 or new fig. 14 localize on fairly similar regions, while apparently detecting different things there. \
> Similarly, the localization of the Doberman features from figures 1 and new fig. 13 could be roughly described as head, chest, left leg, right leg, and left ear, but it is extremely unlikely that a model would actually learn to rely on features at that level of abstraction and redundancy. Instead, some of them will likely focus on e.g. texture rather than the more abstract concept.

---

> ### Author Response · Authors · 2024-11-18
> **Question Regarding Structural Grounding on other Datasets**
>
> > Structure ground is only evaluated on CUB-2011, and the authors also explained why. How much effort would it take to achieve the evaluation metric on a larger dataset, such as ImageNet?
>
> Structural Grounding is evaluated using the class-level annotations that exist for CUB, but not for ImageNet. Evaluating this on ImageNet would require annotations of the human-like similarity between all pairs of classes in ImageNet, which is not available as is. \
> One option to get an estimate for the metric on ImageNet would be to use the similarity of the class names in some language space like CLIPs (https://arxiv.org/abs/2103.00020). \
> This does however have several weaknesses compared to using human annotations. We included this ablation and a discussion of it in new Table 9 and Appendix I. Presumably due to the limitations, QPM is not significantly better than SLDD or Q-SENN, but still far ahead of the baseline Resnet50.

---

> ### Author Response · Authors · 2024-11-18
> **Question Regarding Runtime Compared to Competitors**
>
> > How long is the optimization of “4 hours for fine-grained datasets, and roughly 11 hours for ImageNet”? Are these times better than previous methods?
>
> These times refer to the time it takes to optimize the QP, which is part of our novel method. \
> However, one can compare the time it takes for going from a dense model to a final model, which is something Reviewer GMvn has also asked for. \
> We have added new table 6 and appendix F to present the result. Compared to previous models that follow a similar pipeline, the sparsification and feature selection is significantly accelerated for ImageNet, and slower on datasets with fewer samples. However, note that the method is not optimized for speed as one would only train it sporadically, as the human that follows the decision-making also learns about the class explanations.

---

### Author Response · Authors · 2024-11-18
**Intermediate Revised Manuscript Uploaded**

We thank all reviewers for their valuable and highly constructive feedback. We further appreciate the feedback of a well written manuscript, an intuitive idea, the clever relaxation of constraints, the good accuracy, and following an important goal of global interpretability. \
We respond to all weaknesses / questions separately and want to stress that the proposed additions will improve the overall quality and readability of our manuscript.  \
We uploaded an intermediate version of the paper, as we followed Reviewer GMvn’s suggestion to add more visualizations. Therefore, we also added tables and figures containing results that are referenced in our comments. When referencing parts of the paper, we try to be clear, whether it is in the initial or revised version. References to the revised version will be accompanied by “new”, and else the initial pdf is meant to maintain the references in the comments. \
The changes compared to the initial version are:
* **New Figures 11-14**: Following Reviewer GMvn’s suggestion, we added further example visualizations of our explanations. Fig. 11 contains all the GradCAM visualizations of Fig. 10, showing that QPM consistently localizes on the dog. As QPM enables more in-depth analysis, Fig. 12 and 13 are concerned with Fig.10j, visualizing the features of the predicted Greater Swiss Mountain Dog and the annotated Rottweiler class on class examples and Fig. 10j. Given the class examples and their features, the classification of the presumably falsely labeled image as Greater Swiss Mountain Dog is expected. Fig. 14 then contains exemplary test classifications of the model explained in Fig. 4. As expected by its global explanation, it does not identify a Bronzed Cowbird if the red eye is not visible enough.
* **New Appendix F, Figure 15 and Table 6**: Following questions from Reviewers vzLV and GMvn, we added Results on the runtime of our method with respect to the number of features, as inquired by Reviewer  vzLV and a general comparison and discussion of it.
* **New Table 7, Figure 16 and Appendix G**: We added an Ablation study on the impact of enforcing exactly $n_{wc}$ features per class as opposed to on average, following remarks from Reviewers x2rh and GMvn. The ablation study shows that while being beneficial for interpretability, a balanced assignment also improves accuracy. We further show that the features do indeed become less general during unbalanced fine-tuning in Figure 16.
* **New Figure 17**: The added figure shows the sparsities of the probed models in the above ablation study. It visualizes that Q-SENN learns very unevenly sparse representations. This might explain why SID@5 can be better for it than for QPM, as the top 5 features explain the entire reasoning of QPM, and typically only part of Q-SENNs.
* **New Figure 18 and Appendix H**: In Response to the question regarding polysemantic features, brought up by Reviewer x2rh and also relevant to questions brought up by the other reviewers, we added a discussion of the topic and Figure 18 as anecdotal evidence for the presence of globally polysemantic, but locally monosemantic features. We want to emphasize that the notion of locality used here, is enabled through QPMs interpretable structure.
* **New Table 8**: We added Results on Diversity@5 to the appendix to demonstrate its insufficiency for PIP-Net. As remarked by  Reviewers  vzLV and x2rh, QPM does not perform best in the preferable metric SID@5, so this table can be further context that the relative diversity is  metric dependent, as QPM learns the most diverse features according to  Diversity@5. However, we want to emphasize that a value of around 90% in either of these metrics indicates very localized features and that higher values can also indicate errors, like the class-detectors of PIP-Net.
* **New Table 9 and Appendix I**: In response to the question from Reviewer vzLV, we evaluated Structural Grounding using the similarities in a CLIP Language Space.
* **New Table 10 and Appendix J**: In response to the question from Reviewer vzLV, we compared  AUROC, as alternative highly non-linear assignment criterion, with our linear correlation, showing slight but clear improvements when using the proposed correlation.
* **New Appendix K**: We started the Limitations and Future Work section and included a discussion regarding insertion / deletion metrics for evaluating saliency maps , following remarks of Reviewer x2rh.

Notably, we have not changed the main paper yet. Except for the changes we concretely propose, we will of course refer to all content added to the appendix where suitable in the main paper.  Similarly, the text in the appendix, e.g. in E, is not yet updated with references to the added figures. Finally, the Limitations and Future Work section is not finished yet.

---

### Author Response · Authors · 2024-11-27
**Final Revised Manuscript Uploaded**

We thank all reviewers for their valuable and highly constructive feedback which led to an improved manuscript. We further appreciate the feedback of a well written manuscript, an intuitive idea, the clever relaxation of constraints, the good accuracy, and following an important goal of global interpretability. \
We have now uploaded the revised version, in which we believe we included the results of our discussion. \
The total changes compared to the initial version are:
* All changes described in the comment "Intermediate Revised Manuscript Uploaded"

* **New Appendix K**: We completed the Limitations and Future Work section. It includes:
  * A discussion of high-stakes applications as a valuable avenue for future work, following remarks by Reviewer vzLV.
  * A discussion of negative reasoning as limitation to the QP in its current form, following the comment by Reviewer GMvn.
  * A discussion of polysemantic features and their alignment as current challenge and avenue for future work.
  * A discussion regarding insertion / deletion metrics for evaluating saliency maps, following remarks of Reviewer x2rh.
* **New Lines 153-154**: We added a clarifying sentence to the related work to indicate that our method also profits off of post-hoc alignment methods, following a comment by Reviewer x2rh. This is also discussed in new appendix K.
* **New Lines 125-127**: We clarified the phrasing following the comment from Reviewer GMvn.
* **New Lines 527-529**: We added references to the added ablations
* **New Lines 483-485**: We added references to the added analyses
* **New Line 456**: We added a reference to results on div@5
* **New Line 473**: We reference the runtime analysis in appendix F.
* **Rephrased 4.3, Lines 473-485**: This section is slightly rephrased to put more emphasis on QPM as the only model that excels in all metrics, PIP-Net and glmsaga's weaknesses, and to include references to the appendix where suitable.
* We extended Appendix E to also discuss the new figures 11-14.
* In order to reference the content added to the appendix, we slightly shrank Figure 2 by removing whitespace between the feature images.

---

### Meta-Review · Area_Chair_SCSw · 2024-12-20

**Metareview:**

The paper proposes a method to make image classification interpretable by using a binary assignment of a few features for each class. The method starts with pre-trained network and then imposes these constraints through a quadratic programming framework. This results in a globally interpretable model. Reviewers liked the interpretable framework and that the accuracy did not drop much. Most reviewers also said that the paper was well written and read well. Some of the weakness the reviewers pointed were with using a very few set of features across all classes and perhaps the base model being  normally trained might pose restrictions. The rebuttal seem to address these issues. Overall, the ratings are in favor of acceptance to ICLR.

**Additional Comments On Reviewer Discussion:**

Reviewer x2rh and Reviewer GMvn were happy with the author response and had a favorable rating for the paper. Reviewer vzLV did not respond to the author rebuttal. There were no follow ups on the rebuttal.

---

### Decision · Program_Chairs · 2025-01-22

Accept (Poster)